# Train-Attention: Meta-Learning Where to Focus in Continual Knowledge Learning

**Yeongbin Seo**          **Dongha Lee** *          **Jinyoung Yeo** *
Department of Artificial Intelligence
Yonsei University
{suhcrates,donalee,jinyeo}@yonsei.ac.kr

## Abstract

Previous studies on continual knowledge learning (CKL) in large language models (LLMs) have predominantly focused on approaches such as regularization, architectural modifications, and rehearsal techniques to mitigate catastrophic forgetting. However, these methods naively inherit the inefficiencies of standard training procedures, indiscriminately applying uniform weight across all tokens, which can lead to unnecessary parameter updates and increased forgetting. To address these shortcomings, we propose a novel CKL approach termed Train-Attention-Augmented Language Model (TAALM), which enhances learning efficiency by dynamically predicting and applying weights to tokens based on their usefulness. This method employs a meta-learning framework that optimizes token importance predictions, facilitating targeted knowledge updates and minimizing forgetting. Also, we observe that existing benchmarks do not clearly exhibit the trade-off between learning and retaining, therefore we propose a new benchmark, LAMA-CKL, to address this issue. Through experiments conducted on both newly introduced and established CKL benchmarks, TAALM proves the state-of-the-art performance upon the baselines, and also shows synergistic compatibility when integrated with previous CKL approaches. The code and the dataset will be available online[2]

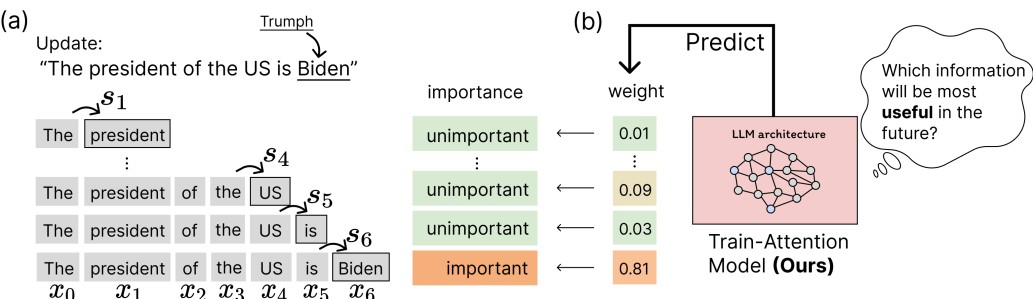

Figure 1: (a) Learning of Causal LM: The document is decomposed into multiple token sequences $s_i \doteq x_i | x_{<i}$[3], which aligns with different importance, but uniformly weighted. (b) Train-Attention: Our proposed Train-Attention learns to predict weights that approximate importance, to enable targeted continual knowledge updates through label-free meta-learning method.

---

\*\*Co-corresponding authors

[2]https://github.com/ybseo-ac/TAALM
[3]In this work, we use the notation $x_i | x_{<i}$ to denote that the element $x_i$ is sequenced after the preceding elements $x_0, x_1, ..., x_{i-1}$.

# 1 Introduction

Large language models (LLMs), pre-trained on extensive text corpora, have demonstrated remarkable effectiveness when fine-tuned or prompted to perform a variety of downstream tasks [Brown et al., 2020, Raffel et al., 2020, Sanh et al., 2021, Wei et al., 2021]. However, as the world changes and new knowledge needs to be updated to the parameters, these models often suffer from a significant loss of previously learned knowledge (i.e., catastrophic forgetting [Kemker et al., 2018, Kirkpatrick et al., 2017]). To address this issue, the field of continual knowledge learning (CKL) is being actively researched [Jang et al., 2021, 2022], which aims to teach a model new knowledge while minimizing forgetting of previous knowledge. Previously explored approaches are broadly categorized into three: (1) minimizing parameter changes through regularization, (2) training the expanded parameters of the adapter while freezing the base model parameters, and (3) reviewing old knowledge. However, these approaches naively inherit the inefficiency of the standard fine-tuning procedure of causal LMs, which uniformly apply weights to all tokens, regardless of their importance.

This inefficiency of uniform weighting becomes more significant within the context of CKL, where the model is assumed to possess a substantial amount of world knowledge and grammatical capabilities already, thus emphasizing the need for limiting targets of learning. For example, consider a causal LM that has undergone both pre-training and fine-tuning and now requires to update the new information that "The president of the US is Biden." Figure 1a illustrates how the model processes the example sentence. The only sequence that carries the essential information of this sentence is the final sequence $s_6$ ("The president of the US is" → "Biden"), which encapsulates the context of "US", "president", and "Biden". Conversely, another sequence such as $s_4$ ("The president of the" → "US") only contains information that is already familiar to the model: the close association between "president" and the name of a nation, as well as the grammatical rule that a noun follows "the". Moreover, $s_1$ ("The" → "president") introduces a harmful bias, suggesting that "president" should invariably follow "The", although any nouns could follow "The". If the model overemphasizes the likelihood of this sequence, several issues can arise: (1) Parameters will be updated more than the necessary amount to learn only essential information, thus resulting in more forgetting. (2) The training steps required to learn the important sequence could become prolonged.

Therefore, we hypothesize that focusing learning efforts on important tokens elevates the performance of the CKL. We present empirical evidence of this in §4.1.1 (paragraph of the analysis on Oracle). The concept of selecting important tokens has been previously explored outside the domain of CKL by Hou et al. [2022], Lin et al. [2024], and demonstrates enhanced performance on downstream tasks. These methods share the same principle, assigning more importance (we denote this "token importance") to the token with higher classification error, which assumes a definition of token importance as "tokens with low-confidence are important". While this approach can accelerate learning of low-confidence tokens, it is still not guaranteed that such low-confidence tokens are truly "important". This emphasizes a need for a more comprehensive definition of "token importance". To clarify this, in the example of Figure 1, it is necessary to consider why human intuition easily accepts that the sequence $s_6$ is more important than others. This understanding comes from the anticipation that knowing the new president will be useful in the future (e.g., conversation with neighbors, school exams, etc) [Land and Furneaux, 1997]. Building on this concept, we define "token importance" as the expected utility of the token in related tasks, a concept we refer to as **usefulness**. Upon this definition of token importance, we propose a novel approach to CKL, named **Train-Attention-Augmented Language Model** (**TAALM**), which predicts weights of each token based on their usefulness, leveraging this weight on the training phase to enable efficient update of new knowledge. Train-Attention, the supportive model that predicts weight for the base model, is trained through the meta-learning method.

We also introduce a new CKL benchmark, **LAMA-CKL**, designed to offer a more clear comparison of learning and retention performance. This benchmark's advantages over the previous standard are explained in §4.3. We experiment on LAMA-CKL and previous CKL benchmark (TemporalWiki [Jang et al., 2022]), and our method achieves remarkable **state-of-the-art** performance on both. Our method is compatible with other approaches, and shows enhanced performance when integrated, indicating a synergistic effect. We also compared RHO-1 [Lin et al., 2024], which is the recent concurrent work on the token selecting method, where ours shows superior performance on CKL benchmarks. Our main contribution can be summarized in three. (1) We propose a novel token weighting approach to the CKL task, with a novel problem definition and meta-learning method.

(2) A new benchmark for CKL based on the LAMA dataset. (3) Through extensive experiments, TAALM proves notable improvements over the baselines.

## 2   Related Works

**Continual Knowledge Learning**   Continual Knowledge Learning (CKL) [Jang et al., 2021] is one variation of Continual Learning (CL), specified to LLM. It is more focused on updating new knowledge without catastrophic forgetting [Kirkpatrick et al., 2017, Kemker et al., 2018] of previously learned and preservable knowledge. Previous approaches for CL and CKL can be mainly categorized into three: regularization, architectural, and rehearsal. We analyze that the three approaches share a common goal; to minimize changes in parameters from initial points. **(1) Regularization**: directly controlling the extent of change in the parameters through weight regulation such as L2 [Kirkpatrick et al., 2017, Zenke et al., 2017, Lopez-Paz and Ranzato, 2017, Aljundi et al., 2018, Chen et al., 2020]. **(2) Architectural**: freezing the base model parameters and expanding learnable parameters with adapters such as Lora [Houlsby et al., 2019, Hu et al., 2021, Wang et al., 2020, Dettmers et al., 2024], thereby keeping initial parameters untouched. **(3) Rehearsal**: method of continually reviewing the data that is employed to train the initial model, ultimately returning the parameters to the initial points [Shin et al., 2017, Sun et al., 2019, He et al., 2019, Rolnick et al., 2019]. In this view, our method is another approach to achieve the same goal, minimizing change of parameters, by filtering objective tokens.

**Meta-Learning**   Meta-learning [Finn et al., 2017, Hospedales et al., 2021] is most commonly understood as learning-to-learn; the process of improving a learning episode, over multiple outer learning episodes. During meta-learning, an outer (i.e., meta) learner is fitted to improve the learning of the inner (i.e., base) model. The meta-learner could be an initial parameter of the base model [Finn et al., 2017], an optimizer of the base model [Andrychowicz et al., 2016], or a hyper-parameter of the base model such as learning-rate [Li et al., 2017, Franceschi et al., 2018]. In this view, our meta-learner (Train-Attention) is an LLM architectural model that predicts hyper-parameters of the base model, as the token weights serve as the hyper-parameters in the training objective.

**Token Selecting**   Methods to enhance learning by selecting specific tokens have previously been explored through various approaches: Token-Dropping [Hou et al., 2022], Focal Loss [Lin et al., 2017], and RHO-1 [Lin et al., 2024]. These methods share a common principle: assigning more importance to the token with higher classification error.

## 3   Train-Attention-Augmented Language Model (TAALM)

### 3.1   Token Importance and Token-Weighted Learning (TWL)

$$\text{PPL}_\theta = -\frac{1}{N} \sum_i \log p(x_i | x_{<i}; \theta) \tag{1}$$

$$= -\frac{1}{\sum_i w_0} \sum_i \log p(x_i | x_{<i}; \theta) \times w_0 \quad (w_0 = 1) \tag{2}$$

$$\text{Token Weighted (TW) PPL}_\theta = -\frac{1}{\sum_i w_i} \sum_i \log p(x_i | x_{<i}; \theta) \times w_i \tag{3}$$

To learn a document data $\mathcal{D} = \{x_1, ..., x_N\}$ which is defined as a series of tokens, a dominant causal language model (LM) ($\theta$) commonly employs perplexity (PPL) as the objective function, as formalized in Eq.(1). This can be also interpreted in the form of Eq.(2), which assigns a uniform weight ($w_0 = 1$) to log probabilities of each sequence $x_i | x_{<i}$ across all documents. In contrast, our proposed methodology assigns weights $0 < w_i \leq 1$ to log probabilities of each sequence, which approximates the importance of each sequence, named token importance [Hou et al., 2022]. We denote this set of weights as the **token weight**, and the training process that incorporates the weights (Eq.(3)) as **Token Weighted Learning (TWL)**.

### 3.2   Train-Attention: Meta-Learning to Predict Token Importance

We suggest defining token importance as **usefulness**, which indicates how much the contained information is useful for solving related tasks in the future. Under this definition, a meta-learning

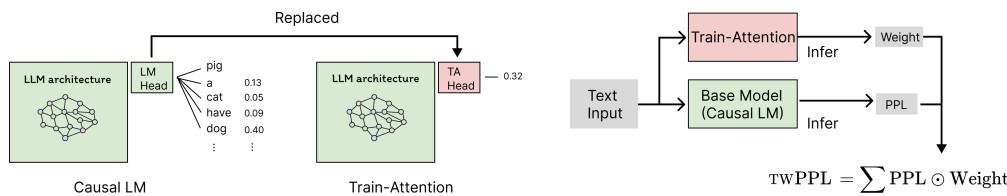

(a) Architecture of Train-Attention                    (b) Train-Attention-Augmented Language Model

$\text{TWPPL} = \sum \text{PPL} \odot \text{Weight}$

Figure 2: (a) depicts the architecture of Train-Attention, which shares the structure of causal LM, while the decoder layer (LM head) of causal LM is replaced from a linear layer of $[hidden\ size \times vocab\ size]$ dimension to $[hidden\ size \times 1]$ dimension, which is TA (Train-Attention) head. (b) depicts the TAALM, where the Train-Attention ($\phi$) is augmented to the base model ($\theta$).

approach can be derived to develop a supportive model (meta-learner) that predicts the optimal token weights. Let $\theta$ represent a base causal LM that continually learns knowledge and solves tasks. $\mathcal{T}_\mathcal{D}$ represents a task that can be solved using information contained in $\mathcal{D}$. Training dataset $\mathscr{D}$ is a set of pairs of $\mathcal{D}$ and $\mathcal{T}_\mathcal{D}$. We assume a task $\mathcal{T}_\mathcal{D}$ can be defined as any type (e.g., predicting object labels, classification) as long as the performance can be measured in a differentiable form. The set of token weights, denoted as $W_\mathcal{D}$, comprises weights $w_i$ that represent the importance of each sequence $x_i|x_{<i}$ within $\mathcal{D}$. The meta-learner, named Train-Attention and denoted as $\phi$, predicts $W_\mathcal{D}$ from $\mathcal{D}$. As illustrated in Figure 2a, $\phi$ inherits the architecture and pretrained parameters of the causal LM, but the decoder layer is adjusted to yield only a single-dimensional float between [0,1] for each position.

The desired process, learning knowledge and solving a task, is described in two steps; (a) *learn*: $\theta$ is trained on $\mathcal{D}$ and is updated to $\theta'$. This update occurs in a TWL manner, with a token weight $W_{\mathcal{D},\phi} \leftarrow \phi(\mathcal{D})$ that $\phi$ predicts upon observing the data $\mathcal{D}$. (b) *solve*: The revised model $\theta'$ is applied to solve the task $\mathcal{T}_\mathcal{D}$, and the loss value $\mathcal{L}_{\theta'}(\mathcal{T}_\mathcal{D})$ is computed to quantify the performance on $\mathcal{T}_\mathcal{D}$, where $\mathcal{L}$ stands for loss function.

Two steps can be regarded as one black box function, which receives $\phi$ as an input and outputs $\mathcal{L}_{\theta'}(\mathcal{T}_\mathcal{D})$. In other words, the task performance of $\theta'$ depends on how $\phi$ gives attention when learning evidence text data. And $\phi$ can be optimized to minimize the $\mathcal{L}_{\theta'}(\mathcal{T}_\mathcal{D})$. For this, the procedure of (a) and (b) is developed to corresponding steps of Eq.(4) and (5), where $\alpha$ and $\beta$ are the learning rates for each respective update.

$$\theta' \leftarrow \theta - \alpha \nabla_\theta \text{TWPPL}_\theta(\mathcal{D}, W_{\mathcal{D},\phi}) \tag{4}$$

$$\phi \leftarrow \phi - \beta \nabla_\phi \mathcal{L}_{\theta'}(\mathcal{T}_\mathcal{D}) \tag{5}$$

---

**Algorithm 1: Optimization of Train-Attention**

---

**Require:** Dataset $\mathscr{D} = \{(\mathcal{D}, \mathcal{T}_\mathcal{D})\}$
**Require:** Learning rate for $\theta, \phi : \alpha, \beta$
**Require:** Max iteration step for training $\theta : M$
    Initialize: base model ($\theta$), Train-Attention ($\phi$)
    **while** $\phi$ not converged **do**
        Sample a data pair $(\mathcal{D}, \mathcal{T}_\mathcal{D}) \sim \mathscr{D}$
        $W_{\mathcal{D},\phi} \leftarrow \phi(\mathcal{D})$          ▷ Predict weights
        **for** $M$ times **do**
            $L_{learn} = \text{TWPPL}_\theta(\mathcal{D}, W_{\mathcal{D},\phi})$
            Evaluate $\nabla_\theta L_{learn}$
            Update: $\theta \leftarrow \theta - \alpha \nabla_\theta L_{learn}$
        **end for**
        ($\theta$ updated to $\theta'$)
        $L_{solve} = \mathcal{L}_{\theta'}(\mathcal{T}_\mathcal{D})$
        Evaluate $\nabla_\phi L_{solve}$
        Update: $\phi \leftarrow \phi - \beta \nabla_\phi L_{solve}$
        Reset $\theta'$ to initial point $\theta$
    **end while**

---

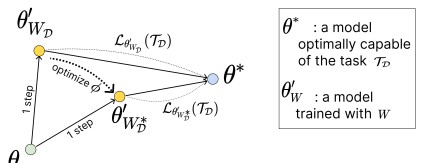

$\theta^*$ : a model optimally capable of the task $\mathcal{T}_\mathcal{D}$

$\theta'_W$ : a model trained with $W$

Figure 3: Optimal $W$ leads $\theta$ closer to $\theta^*$.

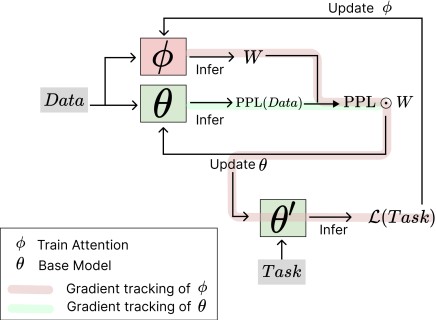

$\phi$ Train Attention
$\theta$ Base Model
     Gradient tracking of $\phi$
     Gradient tracking of $\theta$

Figure 4: One step update of $\phi$.

First, base model $\theta$ is updated to $\theta'$ through TWL, with the token weight $W_{\mathcal{D}}$ which is generated from $\phi$. Second, the meta-learner $\phi$ is updated based on the task performance, $\mathcal{L}_{\theta'}(\mathcal{T}_{\mathcal{D}})$. These two steps of update can be also interpreted like Figure 3. As the model $\theta$ steps out to a new state $\theta'$, the resulting position depends on which token weight ($W$) is applied. Some positions are closer to the $\theta^*$, a model optimally capable of the task $\mathcal{T}_{\mathcal{D}}$, as the distance is measured with $\mathcal{L}_{\theta'}(\mathcal{T}_{\mathcal{D}})$. We can conclude the weight with a shorter distance ($W_{\mathcal{D}}^*$) is more optimal, in the perspective of usefulness. To prevent the $\theta$ from converging to the point $\theta^*$, which disables the measurement of distances, we reset the model parameters to the initial state $\theta$ after every update of $\phi$. More detail is depicted in Figure 4 and Algorithm 1. As the gradients of parameters of $\phi$ are tracked during the updating of $\theta$, its actual implementation is akin to the second derivative. The max iteration step of $\theta$ (denote as $M$ in Algorithm 1) is fixed to 1 through our experiment. We employ gradient accumulation when updating $\phi$ for batch effect.

On the inference phase, $\theta$ learns data in TWL manner as in Eq.(4), with the parameter of $\phi$ frozen. This system is Train-Attention-Augmented Language Model (TAALM). In this work, the foundational structure of $\phi$ is fixed to the small model (TinyLlama-1.1B [Zhang et al., 2024]), while it is augmented to both large (Llama2-7B [Touvron et al., 2023]) and small base models, because $\phi$ is compatible with any base model that shares the same tokenizer. Additionally, we explore utilizing a 101M-parameter bidirectional transformer (BERT) [Devlin, 2018] as a Train-Attention (TA) to further reduce resource requirements.

## 4  Experiment

We conduct experiments on two benchmarks. One is our newly designed LAMA-CKL, and the other is the established benchmark, TEMPORALWIKI [Jang et al., 2022]. We exclude the CKL benchmark by Jang et al. [2021] which is not publicly available. In this section, we present the corpus, evaluation setup, and training detail for Train-Attention and the test result within our proposed LAMA-CKL benchmark. For TEMPORALWIKI, most configurations are aligned with the original work.

### 4.1  LAMA-CKL

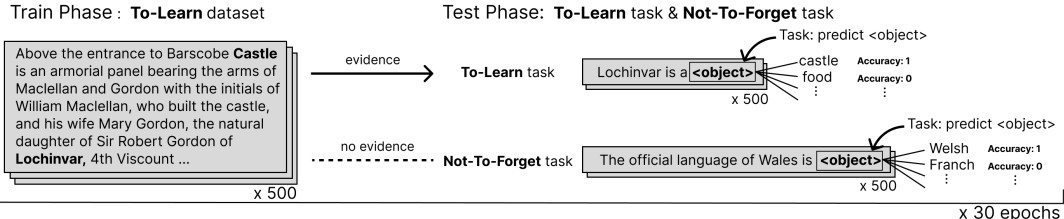

Figure 5: Evaluation procedure of the LAMA-CKL benchmark.

For LAMA-CKL, we tailor the LAMA dataset (LAnguage Model Analysis [Petroni et al., 2019]) to assess the CKL performance, especially the T-REx [Elsahar et al., 2018] part which consists of data from Wikipedia and Wikidata. LAMA is a cluster of datasets that measures how much world knowledge is contained in the LLM. Each unit in the dataset includes a knowledge base triple <subject, relation, object>, along with corresponding evidence documents that support the information contained in this triple. Referring to the previous work [Jang et al., 2022], a CKL benchmark should evaluate both **plasticity** and **stability**. Plasticity refers to how well the model updates new knowledge, while stability refers to how little the model forgets existing knowledge. Accordingly, we sample 500 units of TO-LEARN and NOT-TO-FORGET sets from LAMA to assess each dimension. During the evaluation, as illustrated on Figure 5, the model learns the evidence documents in the TO-LEARN set. It is then tested on both the TO-LEARN task and the NOT-TO-FORGET task to assess plasticity and stability, respectively.

**Dataset Setup**  Here, we outline a protocol for sampling test corpora for the LAMA-CKL benchmark. As TO-LEARN set represents "the knowledge that the model either encounters for the first time or needs to update", it is selected based on two constraints: (1) sample from the categories of time-variant relations, predicated on the assumption that knowledge categorized as time-variant

typically requires updates. (2) to ensure the concept of "knowledge new to the model", we select units where the task accuracy is zero when measured with pre-update baselines. Conversely, because NOT-TO-FORGET set represents "the knowledge that the model already knows and aims to retain", it is selected from categories of time-invariant relations, with task accuracy of 1. We recommend sampling a new dataset by the specified constraints when evaluating models outside of the LLaMA-family, for more accurate assessment. The categorization of time-variant and time-invariant follows Jang et al. [2021]. Each selected unit includes (1) an evidence document, (2) a knowledge base triple (e.g., <Lochinvar, is an instance of, castle>), and (3) a descriptive sentence encapsulating the triple (e.g., "Lochinvar is a castle"), which is inherited from LAMA dataset. The task is predicting object label tokens in the descriptive sentence. Details on data are in Appendix A.1

**Evaluation Setup** During the evaluation, each epoch consists of both a training phase and a test phase. In the training phase, the model is trained on a set of 500 evidence documents from the TO-LEARN set. Subsequently, in the test phase, the model's prediction accuracy for the object labels is assessed using 500 descriptive sentences from both the TO-LEARN and NOT-TO-FORGET sets. This process is repeated over 30 epochs. For the TO-LEARN set, an increase in mean accuracy from 0 signifies the model's plasticity. Conversely, a decline in mean accuracy for the NOT-TO-FORGET set from 1 to lower values indicates the model's stability, as it tends to forget previously learned information.

In the proposed benchmark LAMA-CKL, we suggest four main factors as evaluation indicators. **1) Top Acc:** the highest TO-LEARN accuracy among checkpoints of 30 epoch. **2) Epoch:** the epoch where the Top Acc appears. **3) NF Acc:** NOT-TO-FORGET accuracy of the checkpoint model which is the same as Top Acc. **4) Total Knowledge**: the sum of Top Acc and NF Acc, indicating total capacity of knowledge including updating and maintaining. We chose these factors because the best CKL system is one that *learns the most and the fastest and loses the least*. Factor 1, 3, and 4 are better if higher, while factor 2 is better if lower. The detailed configurations for training datasets are in Appendix A.1.1.

**Train-Attention Training Setup** Referring to Algorithm 1, the training procedure of Train-Attention requires data $\mathcal{D}$ and related task $\mathcal{T}_\mathcal{D}$. For LAMA-CKL dataset, we assign evidence document of each unit as $\mathcal{D}$, and knowledge base triple in a document of **schematic form** as $\mathcal{T}_\mathcal{D}$. The perplexity of **object** token is assigned as the objective of $\phi$. Figure 6 shows the heat map of token-weight that Train-Attention generates. Train-Attention seems to give more attention to entities of certain categories, rather than words with general grammatical roles. We describe the detailed configuration and findings on the training of Train-Attention in Appendix A.2.

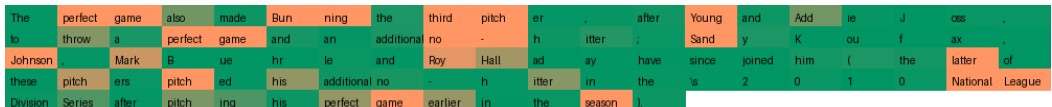

Figure 6: Heat map of token weights from Train-Attention. Orange color indicates higher weights.

**Baseline Setup** We utilize Llama2-7B integrated with QLoRA [Dettmers et al., 2024] as a base model. The baseline methods and their hyper-parameter settings follow previous CKL study of Jang et al. [2022]: standard finetune, K-Adapter [Wang et al., 2020], Mix-review [He et al., 2019], LoRA [Hu et al., 2021], RecAdam [Chen et al., 2020]. We regard standard finetune on QLoRA as a substitute for full finetuning and LoRA, thus skipping the two baselines. We also compare RHO-1 [Lin et al., 2024], which is the most recent concurrent work on the token selecting method, sharing a similar concept with ours. We chose the initial parameter state as a reference model, which is utilized to select important tokens for RHO-1, with other hyper-parameters following the optimal of the original. We also evaluate a model trained in TWL manner with **Oracle** token weight. For which, a weight of 1 is exclusively assigned to the **object** label token in the evidence document, and the rest is assigned zero weight. Oracle is compared for two purposes: **(1)** To prove the concept that token-weighted learning has the advantage for CKL. **(2)** To check the performance upper bound of Train-Attention. Detailed configurations are in Appendix A.3

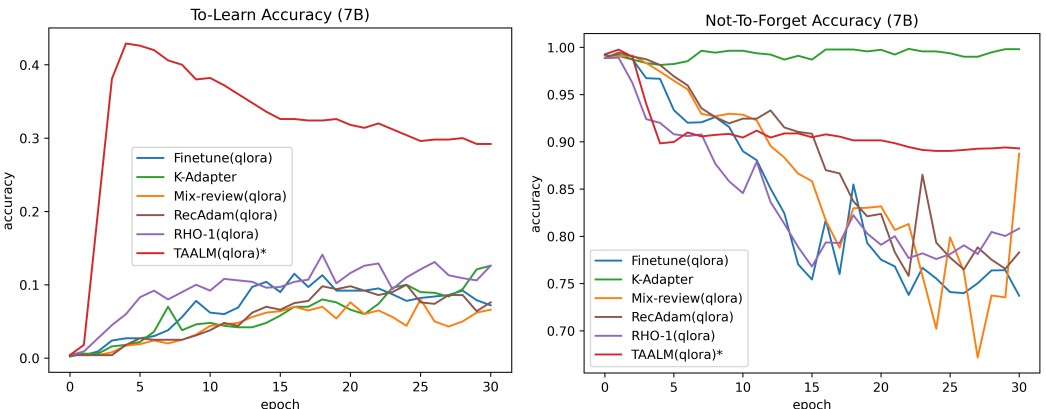

Figure 7: LAMA-CKL performance of large (Llama2-7B) baseline models. The graph on the left represents TO-LEARN task, and the graph on the right represents NOT-TO-FORGET task performance. The x-axis is the learning epoch, and the y-axis is accuracy.

Table 1: LAMA-CKL performance of Llama2-7B based baselines. The evaluation indicator of each column is explained on §4.1. The best performance is marked as **bold** while the second best is underlined.

|  | Top Acc | Epoch | NF Acc | Total Knowledge |
|---|---|---|---|---|
| Finetune(QLoRA) | 0.1150 | 16 | 0.8174 | 0.9324 |
| K-Adapter | 0.1260 | 30 | **0.9980** | 1.1240 |
| Mix-review(QLoRA) | 0.0800 | 25 | 0.7988 | 0.8788 |
| RecAdam(QLoRA) | 0.1000 | 24 | 0.7933 | 0.8933 |
| RHO-1(QLoRA) | 0.1410 | 18 | 0.8223 | 0.9633 |
| TAALM(QLoRA) | **0.4290** | **4** | 0.8983 | **1.3273** |

### 4.1.1 Result & Analysis

**TAALM achieves substantial CKL performance**   As results show in Figure 7 and Table 1, our method (TAALM) overwhelms other baselines on ability and speed of learning. Ours records the Top Acc of TO-LEARN task as 0.4290 on only 4 epochs of updating. The accuracy record of ours is 3.04 times higher than the second place (RHO-1). And the required epoch is only 25% of the second place (standard finetune).

K-Adapter shows the highest NF Acc, rarely forgetting previous knowledge. However, the TO-LEARN accuracy also barely increases, indicating that parameter updates rarely occur. This could be due to the architectural difference from QLoRA, the backbone of other baselines. Therefore, we conclude that it is meaningless to compare K-Adapter and QLoRA based baselines in the same condition. When comparing only the QLoRA base methods, TAALM shows overwhelming performance in all dimensions. Ours learn the most, the fastest, and forget the least. RHO-1, one of the token selecting methods, also shows a bigger capacity for learning and less forgetting than standard finetuning, However, the advantage is minimal compared to ours.

**Combination with TAALM improves all of the previous baselines**   As Train-Attention is an approach to manipulating loss values on the end side, it is easily compatible with previous methods based on other concepts. Thus we combine various methods to TAALM and observe their performance. Referring to the experimental results and details in Appendix A.4.2, each combined version shows a highly improved capacity than the baseline alone. Especially, combining our method with K-Adapter demonstrates a considerable balance between stability and plasticity.

**Oracle vs Train-Attention vs Standard finetune**   It is assumed that TWL with Oracle label is the upper bound of performance of TWL with Train-Attention, because it is sharply targeting only

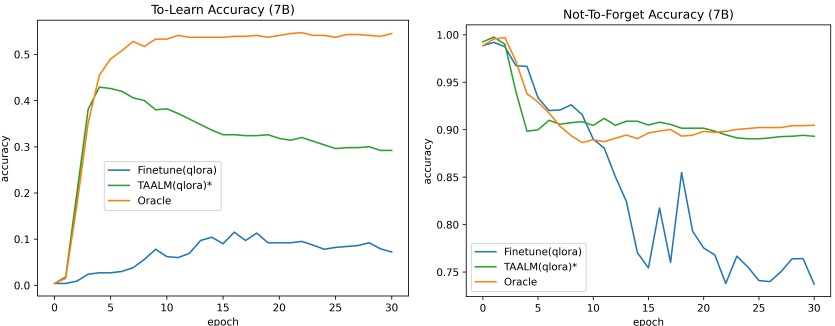

Figure 8: Comparison between Oracle, standard finetuning, and ours, tested on LAMA-CKL.

the necessary token. Referring to Figure 17, for Oracle, Top Acc is 0.5470, epoch 17, and NF Acc is 0.9002. Oracle shows that Top Acc is 4.75 times higher and NF Acc is 1.11 times higher than standard finetune, thus proving the substantial advantage of token-weighted learning on CKL. Also, TAALM nearly approaches Oracle, as it achieves 78.2% of Oracle Top Acc. NF Acc of ours also maintains a similar level to Oracle, indicating that Train-Attention is optimized close to the upper bound. It also indicates that optimization through meta-learning could excel human labeled weight.

**Small (1B) TAALM excels large baselines**   We also experiment on the smaller (TinyLlama-1B) baselines, and TAALM on 1B records the best compared to 7B baselines. This observation indicates that our method outperforms other baseline methods even with significantly smaller parameter sizes and computational resources. Related Table and Figure are on Appendix A.4.1.

**TA on BERT achieves comparable performance with small resources**   We train BERT as a TA and evaluated it on the LAMA-CKL dataset. Training TA on BERT requires only a single 24GB GPU, significantly reducing resource usage compared to the previous model (single 82GB GPU), yet achieving performance similar to the larger (1.1B) TA (Appendix D).

**Ablation study on various design choices**   We conduct an ablation study on various design choices applied to the token importance predicted by TA: (1) masking out tokens (setting importance to 0) in real-time when the prediction matches the label, and (2) dropping weights with token importance below the top k% threshold. The ablation study reveals that heuristic adjustments degrade performance, as TA is already in an optimized state. Details are in Appendix E.

## 4.2   TEMPORALWIKI

We experiment on the original CKL learning benchmark TEMPORALWIKI [Jang et al., 2022], where models have to continually learn Wikipedia documents of serial periods (0809, 0910, 1011, 1112) and test on the corresponding TWIKI-PROBES, which is a dataset of knowledge base triples. As we train Train-Attention on the 0809 data, tests are conducted on the rest. We experiment with only a small (TinyLlama-1B) model, which is bigger than the baselines of the original work (GPT-2 Large). We conduct a separate experiment on QLoRA-based K-Adapter based models, referring to the analysis on experiment of LAMA-CKL on §4.1.1. We only consider training DIFFSET, which is the only changed part of Wikipedia, because it is reported as a condition of the best performance. Most of the experimental settings follow the original, and additional change is described in Appendix C.

### 4.2.1   Result & Analysis

Referring to Table 2, our method presents the state-of-the-art performance across both experiments on QLoRA based and K-Adapter based models. This achievement is consistent in all periods, and in both CHANGED and UNCHANGED TWIKI-PROBES. This result is aligns with the LAMA-CKL benchmark result, showing that our method has a substantial advantage on the CKL. QLoRA based baselines showed poor performance compared to K-Adapter based baselines, indicating architectural disadvantage. We also test TAALM optimized for LAMA-CKL on the TEMPORALWIKI, referring

Table 2: TEMPORALWIKI performacne of small (TinyLlama-1B) baselines. **Un** refers UNCHANGED, **C** refers CHANGED, **Avg** refers the average of the two. TAALM is our method.

(a) QLoRA based

| | TWiki-Probes-0910 | | | TWiki-Probes-1011 | | | TWiki-Probes-1112 | | |
|---|---|---|---|---|---|---|---|---|---|
| | **Un** | **C** | **Avg** | **Un** | **C** | **Avg** | **Un** | **C** | **Avg** |
| Finetune(QLoRA) | 9.999 | 10.057 | 10.028 | 9.554 | 9.531 | 9.543 | 9.736 | 9.632 | 9.684 |
| Mix-review(QLoRA) | 9.529 | 9.579 | 9.554 | 9.514 | 9.486 | 9.501 | 9.562 | 9.452 | 9.507 |
| RecAdam(QLoRA) | 9.514 | 9.604 | 9.559 | 8.992 | 9.031 | 9.012 | 9.579 | 9.479 | 9.529 |
| RHO-1(QLoRA) | 4.389 | 4.624 | 4.507 | 4.360 | 4.395 | 4.3775 | 4.471 | 4.717 | 4.594 |
| TAALM(QLoRA) | **4.019** | **4.268** | **4.1435** | **4.030** | **4.154** | **4.092** | **4.036** | **4.357** | **4.197** |

(b) K-Adapter based

| | TWiki-Probes-0910 | | | TWiki-Probes-1011 | | | TWiki-Probes-1112 | | |
|---|---|---|---|---|---|---|---|---|---|
| | **Un** | **C** | **Avg** | **Un** | **C** | **Avg** | **Un** | **C** | **Avg** |
| Finetune(K-Adapter) | 2.768 | 2.982 | 2.875 | 2.598 | 2.679 | 2.639 | 2.552 | 2.669 | 2.611 |
| Mix-review(K-Adapter) | 2.486 | 2.724 | 2.605 | 2.334 | 2.446 | 2.390 | 2.284 | 2.425 | 2.355 |
| RecAdam(K-Adapter) | 2.494 | 2.710 | 2.602 | 2.323 | 2.415 | 2.369 | 2.248 | 2.375 | 2.312 |
| RHO-1(K-Adapter) | 2.722 | 2.950 | 2.836 | 2.538 | 2.634 | 2.586 | 2.478 | 2.603 | 2.541 |
| TAALM (K-Adapter) on LAMA-CKL | 2.364 | 2.601 | 2.483 | 2.168 | 2.282 | 2.225 | 2.123 | 2.307 | 2.215 |
| TAALM (K-Adapter) | **1.980** | **2.194** | **1.9705** | **1.907** | **2.034** | **2.087** | **1.901** | **2.070** | **1.986** |

Table 2b. It achieves the second-best performance, indicating robustness across different distributions of tasks.

## 4.3 Why LAMA-CKL: clear contrast of plasticity and stability

For the benchmark TEMPORALWIKI, because DIFFSET is corpora of evidence documents for CHANGED set, learning of DIFFSETS is supposed to result in performance improvement over CHANGED set and forgetting of UNCHANGED set. However, during our experiments, we observe that both CHANGED and UNCHANGED performance tend to move in similar directions when learning DIFFSET, which is in contradiction to our assumption (Appendix C.4). We analyze this for two primary reasons. First, the DIFFSET contains evidence documents for both the CHANGED and the UNCHANGED sets (Appendix C.4). This is a complicating factor in the evaluation of stability. Second, the experimental setup involves training on a vast amount of data, an average of 707K documents per period, for just a single epoch at a low learning rate. This might result in learning little amount of knowledge, while the task ability is challenged by extensive iterations of updates; which is closer to a continual learning setup rather than CKL. To address this issue, we structured the LAMA-CKL as follows: (1) To mitigate the issue of data overlap, we partition the dataset into variant and invariant subsets. These subsets are further classified based on the task accuracy measured by pre-update baselines. (2) We conduct training over multiple epochs on a relatively small dataset to observe the acquirement of knowledge. In practice, our benchmark shows a clear upward trend in the TO-LEARN set and a distinct decline in the NOT-TO-FORGET set as training progresses, clearly demonstrating the contrast between plasticity and stability.

## 5 Conclusion and Limitation

In this paper, we demonstrate that the application of Train-Attention significantly enhances CKL performance and is also synergistic with other baselines. Nevertheless, our work has the following limitations and potential for future exploration.

**Task specificity of Train-Attention** Train-Attention is trained to focus on information related to tasks encountered in the training session. This allows task performance to increase, but on the other hand, it left questions as to whether it would be possible to cope with other tasks. Nonetheless, if the task entails the acquisition of general knowledge, it will be transferable to other tasks sharing similar distributions. For instance, TAALM optimized to LAMA-CKL also achieved the best performance

on the TEMPORALWIKI (§4.2.1). Additionally, Train-Attention can ever evolve to adapt, enabling optimal performance for the current tasks.

**What if there are no data-task pair**    Train-Attention can be trained only if there is a data-task pair. If there is no paired dataset, it can be doubted that training is difficult. However, every knowledge has its purpose, and we can find a strategy to discover it. **1) Search:** When the data and task pools are separate, we can join highly related pairs via searching. TEMPORALWIKI is also a dataset in which data and tasks are not paired, thus we conduct a lexical search. In the future, also dense research methods can be explored. **2) Generate:** If there is even no separate task pool, we can at least get prior information about what kind of tasks are probable to come in the future. Synthetic tasks can be generated via instruction-tuned LLM, based on this prior information. These methods also resemble the human's cognitive strategy, who often revisit past memories and pose hypothetical questions to themselves to enhance the efficiency of their learning processes.

**Broader impacts**    Our method aims to increase the ability of CKL, therefore enhancing the practicability of LLMs and saving the computational resources for fine-tuning entire huge LLMs. We believe that this paper does not have any immediate negative societal impact.

## Acknowledgement

This work was supported by STEAM R&D Project, NRF, Korea (RS-2024-00454458) and Institute of Information & Communications Technology Planning & Evaluation (IITP) grant funded by the Korean government (MSIT)(No.RS-2020-II201361, Artificial Intelligence Graduate School Program (Yonsei University)) and (2022-0-00077, RS-2022-II220077,AI Technology Development for Commonsense Extraction, Reasoning, and Inference from Heterogeneous Data). Jinyoung Yeo and Dongha Lee are the co-corresponding authors.

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

# A    LAMA-CKL Benchmark Additional Detail

## A.1    Dataset setup detail

As each unit of the LAMA dataset contains multiple evidence documents, we sample one document for each unit, based on the following criteria: (1) The document should exceed a length of 70 tokens. (2) The document must include both the subject and object entities. These documents are then truncated to a maximum length of 512 tokens. We assess the accuracy of the knowledge base triple of each unit to filter them. We utilize the TinyLlama-1.1B Zhang et al. [2024] and Llama2-7B Touvron et al. [2023] models, integrated with QLoRA Dettmers et al. [2024] and K-Adapter Wang et al. [2020], total 4 variations. Units with unanimous accuracy among the four models are only selected for TO-LEARN or NOT-TO-FORGET set. Because NOT-TO-FORGET accuracy differs among descriptive and schematic form queries, NOT-TO-FORGET set for descriptive and schematic tasks are separately sampled, allowing overlap. The distribution of relation categories among TO-LEARN and NOT-TO-FORGET is depicted in the Figure 9. Also the relation categories of each TO-LEARN and NOT-TO-FORGET are detailed in Table 4. Table 3 presents data statistics for NOT-TO-FORGET and TO-LEARN sets.

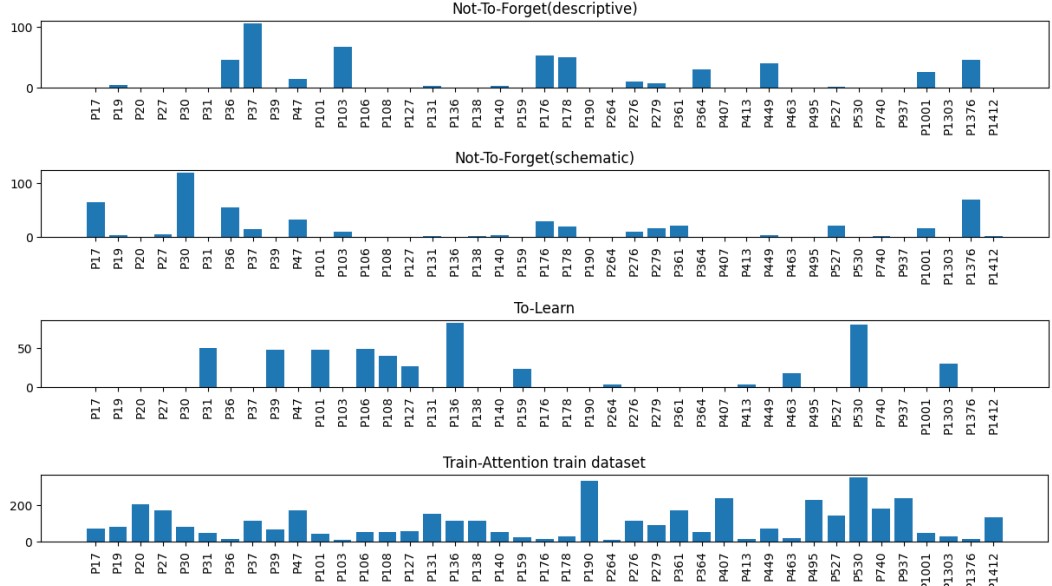

Figure 9: Distribution of relation categories of each dataset in LAMA-CKL

Table 3: LAMA-CKL data statistics.

|  | Size | Avg. Evidence token | Avg. Label token |
|---|---|---|---|
| NOT-TO-FORGET (descriptive) | 500 | 141.1 | 1.5 |
| NOT-TO-FORGET (schematic) | 500 | 153.4 | 1.9 |
| TO-LEARN | 500 | 112.8 | 1.0 |
| Train-Attention Train Data | 4166 | 136.0 | 1.1 |

---

[4]Directly referred from 'Relations of InvariantLAMA of Jang et al. [2021]

Table 4: Relations of each NOT-TO-FORGET and TO-LEARN set.

(a) Relations of NOT-TO-FORGET[4]

| Relation Code | Template ([X], [Y]) | Relation |
|---|---|---|
| P19 | [X] was born in [Y] . | place of birth |
| P20 | [X] died in [Y] . | place of death |
| P279 | [X] is a subclass of [Y]. | subclass of |
| P37 | The official language of [X] is [Y]. | official language |
| P449 | [X] was originally aired on [Y] . | original network |
| P47 | [X] shares border with [Y] . | shares border with |
| P138 | [X] is named after [Y] . | named after |
| P364 | The original language of [X] is [Y] . | original language of film or TV show |
| P527 | [X] consists of [Y] . | has part |
| P176 | [X] is produced by [Y] . | manufacturer |
| P27 | [X] is [Y] citizen . | country of citizenship |
| P407 | [X] was written in [Y] . | language of work or name |
| P30 | [X] is located in [Y] . | continent |
| P178 | [X] is developed by [Y]. | developer |
| P1376 | [X] is the capital of [Y], | capital of |
| P131 | [X] is located in [Y] . | located in the administrative territorial entity |
| P1412 | [X] used to communicate in [Y]. | languages spoken, written or signed |
| P17 | [X] is located in [Y] . | country |
| P276 | [X] is located in [Y] . | location |
| P937 | [X] used to work in [Y]. | work location |
| P140 | [X] is affiliated with the [Y] religion . | religion |
| P103 | The native language of [X] is [Y] . | native language |
| P190 | [X] and [Y] are twin cities . | twinned administrative body |
| P1001 | [X] is a legal term in [Y] . | applies to jurisdiction |
| P495 | [X] was created in [Y] . | country of origin |
| P36 | The capital of [X] is [Y] . | capital |
| P740 | [X] was founded in [Y]. | location of formation |
| P361 | [X] is part of [Y] . | part of |

(b) Relations of TO-LEARN

| Relation Code | Template ([X], [Y]) | Relation |
|---|---|---|
| P31 | [X] is a [Y] . | instance of |
| P39 | [X] has the position of [Y] . | position held |
| P101 | [X] works in the field of [Y]. | field of work |
| P106 | [X] is a [Y] by profession. | occupation |
| P108 | [X] works for [Y] . | employer |
| P127 | [X] is owned by [Y] . | owned by |
| P136 | [X] plays [Y] music . | genre |
| P159 | The headquarter of [X] is in [Y] . | headquarters location |
| P264 | [X] is represented by music label [Y]. | record label |
| P413 | [X] plays in [Y] position . | position played on team / speciality |
| P463 | [X] is a member of [Y] . | member of |
| P530 | [X] maintains diplomatic relations with [Y] . | diplomatic relation |
| P1303 | [X] plays [Y] . | instrument |

### A.1.1 Evaluation setup detail

**Hardware and hyper-parameters** During training TO-LEARN documents, 8 RTX 3090 GPU (24GB) are used, with a global batch size of 64. A total of 30 epochs took 25 minutes of GPU time. Learning rate 1e-4, AdamW optimizer, and max length of 512 tokens are applied.

**Accuracy measurement** Accuracy is utilized as a metric (Top Acc, NF Acc) to assess the efficacy of models in the task of label prediction. This metric quantifies the proportion of label tokens correctly identified by the model out of the total label tokens presented.

## A.2 Train-Attention training detail

**Dataset** We select LAMA units from both time-variant and time-invariant sets, specifically those with an accuracy below 0.5, while ensuring no overlap with the NOT-TO-FORGET and TO-LEARN sets. Furthermore, the initial data exhibit a disproportionately large number of units categorized under relation P530 ('diplomatic relations of the country'). To address this imbalance, we adjust the frequency of P530 units to match that of the second most prevalent type. Consequently, the final version of the Train-Attention training dataset comprises a total of 4166 units, detailed in Table 3.

We propose the **schematic form** to arrange the knowledge base triple into a human-readable sentence. It fits the one-directional feature of causal LM, as opposed to the descriptive form where important information sometimes appears behind the label tokens. The template and example are on Appendix B. We employ schematic form in training Train-Attention, while employing descriptive form during evaluation.

**Training** We utilized small (TinyLlama-1B) and large (Llama2-7B) models from the LLaMA family, integrated with QLoRA [Dettmers et al., 2024]. We found that using a large rather than small model for the base model ($\theta$) results in faster convergence. On the other hand, employing a large or a small model for the Train-Attention model ($\phi$) made little difference in the aspect of the convergence step and validation score. So we adopt a large model for $\theta$ and a small model for $\phi$ while training Train-Attention. In the test phase, We find that $\theta$ and $\phi$ are still compatible even though they don't share the same background model, as long as they use the same tokenizer. Therefore we utilize small-size Train-Attention for both small and large baseline experiments.

We initialize the parameters of the Train-Attention head, which is a decoder layer as depicted in Figure 2a, using a normal distribution with a mean of 0 and a standard deviation of 0.0001. Consequently, the weights generated by the initialized Train-Attention are equivalent to the uniform weight where all $w_i = 1$. This approach is adopted to observe the convergence of the $\mathcal{L}_{\theta'}(\mathcal{T}_{\mathcal{D}})$ more clearly, ensuring that the loss value declines from the initial state if training proceeds normally.

A single A100 (82GB) GPU is used, and the effect of batch size 16 is achieved through gradient accumulation. A checkpoint of 400 global steps is used (takes about 6 GPU hours). Of the 4166 train data, 100 are used for validation. The learning rate of 2e-4 and AdamW optimizer is employed for both the base model and Train-Attention.

## A.3 Baselines detail

In this section, we describe the detailed configuration of QLoRA and K-Adapter. The other baselines follow the previous CKL work of [Jang et al., 2022].

**QLoRA** Hyper-parameters of QLoRA follow one of the optimal of the original work [Dettmers et al., 2024]. We employ LoRA $r = 64, \alpha = 16$, NF4 with BF16 computation datatype. A total of 160M parameters are expanded for the large (Llama2-7B) baselines.

**K-Adapter** Hyper-parameters of K-Adapter follow the previous CKL work of [Jang et al., 2022]. A total of 303M parameters are expanded. The trainable parameters are double the QLoRA. Efforts are made to reduce the learnable parameters of K-Adapter, but we observe that the current settings are the minimum level necessary for K-Adapter to function effectively. In experiments, K-Adapter demonstrates greater stability than QLoRA, which could be partially attributed to its larger parameter size. We also employ the same quantization configuration of QLoRA for K-Adapter, NF4 with BF16 computation datatype.

### A.4 Experimental results additional detail

#### A.4.1 LAMA-CKL Experiment on small (1B) baselines

Figure 10 and Table 5 describe performances of small (TinyLlama-1B) baselines on the LAMA-CKL. The performance of TAALM based on the small (TinyLlama-1B) architecture is significantly better even when compared to the large (Llama2-7B) baselines.

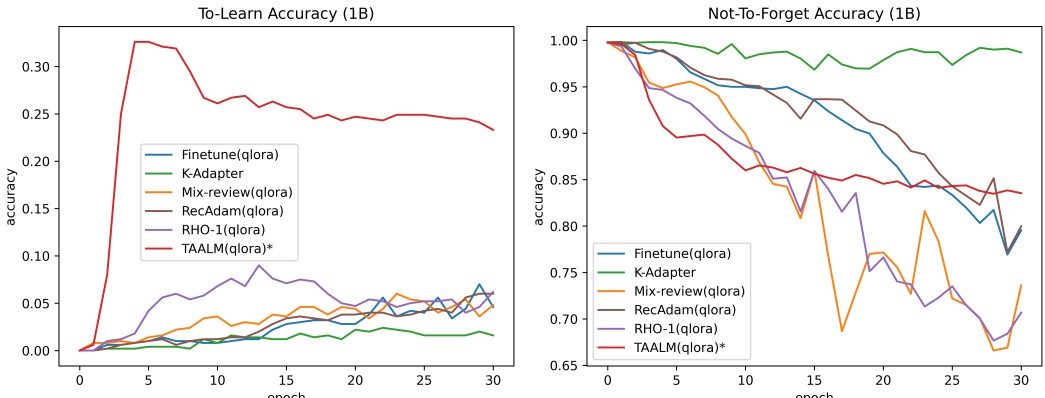

Figure 10: LAMA-CKL performance of small (TinyLlama-1B) baselines.

Table 5: LAMA-CKL performance of small (TinyLlama-1B) baselines.

|                    | Top Acc    | Epoch  | NF Acc     | Total Knowledge |
|--------------------|------------|--------|------------|-----------------|
| Finetune(QLoRA)    | 0.0700     | 29     | 0.7693     | 0.8393          |
| K-Adapter          | 0.0240     | 22     | **0.9908** | 1.0148          |
| Mix-review(QLoRA)  | 0.0600     | 23     | 0.8161     | 0.8761          |
| RecAdam(QLoRA)     | 0.0600     | 29     | 0.7719     | 0.8319          |
| RHO-1(QLoRA)       | 0.0900     | 13     | 0.8523     | 0.9423          |
| TAALM(QLoRA)       | **0.3260** | **4**  | 0.9078     | **1.2338**      |

#### A.4.2 Combination of ours (TAALM) and other baselines

Figure 11 presents that the performances of all baselines are significantly improved when combined with the Train-Attention, indicating synergistic compatibility of our method.

Referring to Table 6 and Figure 12, the combination of ours and Mix-review shows the highest Top Acc and Total Knowledge among all variations. Unlike Mix-review alone, the Mix-review combined with ours applies Train-Attention not only to the train data but also to the review data. As a result, it is observed that the training becomes more stable than the original, which tends to be unstable due to the doubled amount of data (train + review). This indicates that the combination of Train-Attention with the rehearsal approach is synergistic. Variations of "ours + K-Adapter" show relatively low learning capability (Top Acc, Total Knowledge) while featuring still robust stability. Therefore, a combination of ours and K-Adapter can be considered a balanced variation between learning and maintaining.

Figure 13 presents the trade-off between plasticity (TO-LEARN) and stability (NOT-TO-FORGET) of all baselines, including all variations of combination. Variations combined with our method are exclusively positioned in the upper right quadrant, indicating a minimal trade-off between updating and forgetting.

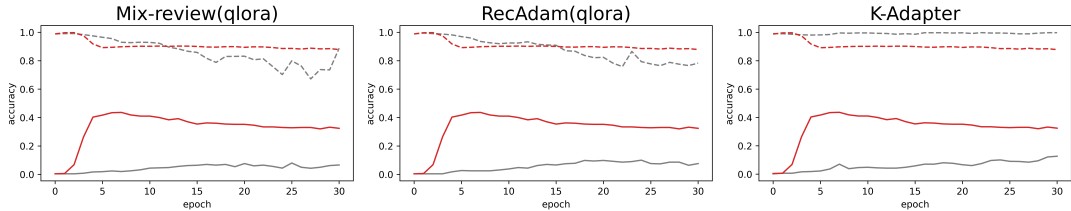

Figure 11: Comparison of each baseline alone and combined with our method. Each title on the plot represents the baseline method. The **gray line** represents the baseline alone, and the **red line** represents the combination with TAALM. Solid line for TO-LEARN, dashed line for NOT-TO-FORGET. All are based on Llama2-7B, and tested on LAMA-CKL.

Table 6: Combination of ours (TAALM) and other baselines. Based on Llama2-7B, tested on LAMA-CKL.

|  | Top Acc | Epoch | NF Acc | Total Knowledge |
|---|---|---|---|---|
| ours + K-Adapter | 0.3320 | 21 | **0.9747** | 1.3067 |
| ours(QLoRA) + Mix-review | **0.4500** | **3** | 0.9012 | **1.3512** |
| ours(QLoRA) + RecAdam | 0.4360 | 7 | 0.8982 | 1.3342 |
| ours + K-Adapter + RecAdam | 0.2640 | 13 | 0.9730 | 1.2370 |
| ours + K-Adapter + Mix-review + RecAdam | 0.3140 | 25 | 0.9677 | 1.2817 |
| ours(QLoRA) | 0.4290 | 4 | 0.8983 | 1.3273 |

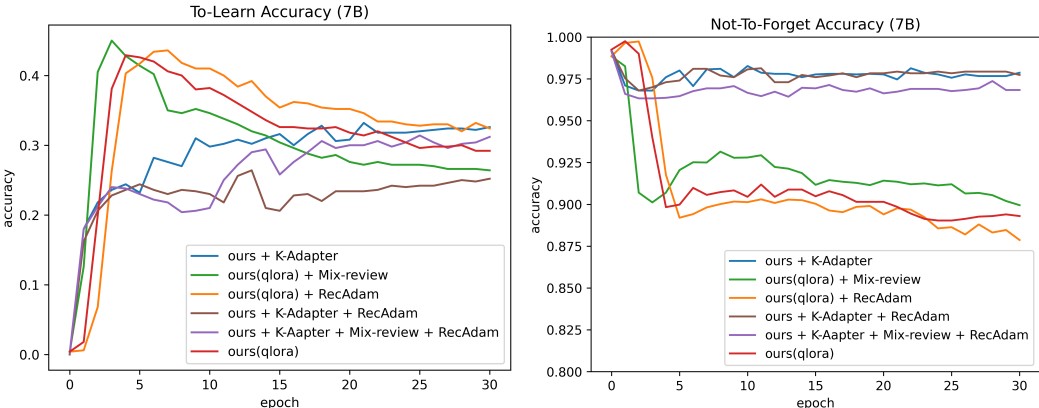

Figure 12: Combinations of other methods with ours (TAALM) on Llama2-7B base model, tested on LAMA-CKL.

### A.4.3 Rationale for the accuracy drop

In the main experiment (Figure 7), the accuracy declines after a rapid peak after about the fourth epoch. We hypothesize this is a result of overfitting. While the token weights from TA include beneficial targets, there must also be some that are not. Performance achieves peak until the model completes learning for the true target, but learning may continue for the false targets. This continued learning leads to parameter updates in suboptimal directions, resulting in forgetting.

The comparison between TA and the oracle (Figure 17) provides evidence for this rationale. As the oracle's performance does not decline with further training, the differences in accuracy trends could be caused by false targets. This phenomenon can also be interpreted as a type of overfitting where the model becomes too fitted to the training data which has different distribution from test data. As the phenomenon of declining after peaking is common due to overfitting, this cycle seems to occur more rapidly for TAALM.

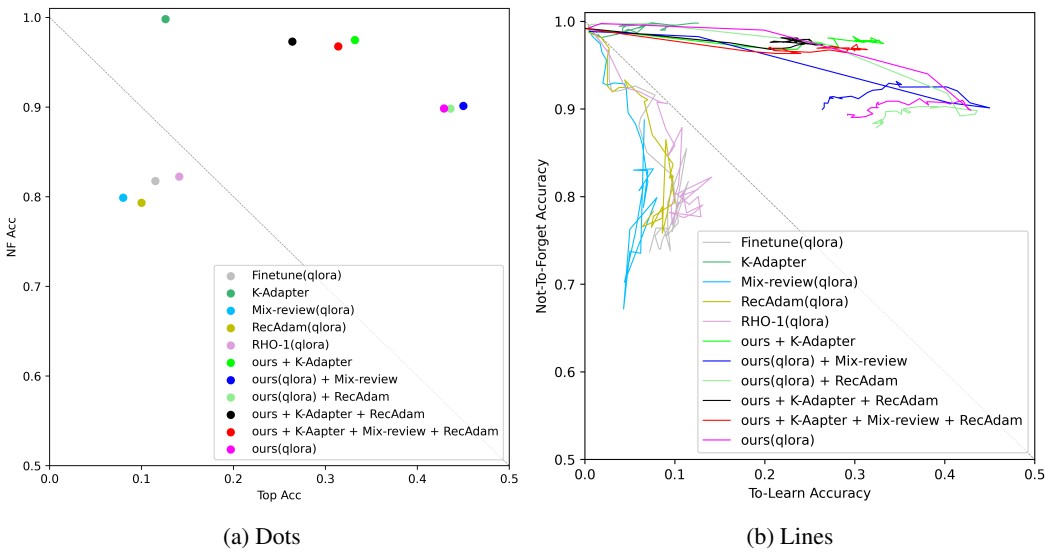

(a) Dots

(b) Lines

Figure 13: The trade-off between plasticity (TO-LEARN) and stability (NOT-TO-FORGET) are visualized, for all baselines including combinations with our method. All models are based on large (Llama2-7B) architecture. Gray dashed lines stand for the zero-sum state where the Total Knowledge (sum of the performances of TO-LEARN and NOT-TO-FORGET) is 1. The right and upper sides of the gray lines indicate the more efficient system where learning causes less forgetting. **(a)Dots** presents only the checkpoints of the Top Acc, while **(b)Lines** presents whole checkpoints of 30 epochs as lines.

#### A.4.4 Reporting error ranges of the main experiment

We conduct five independent runs using different random seeds on the training data-loader, and display the results with error ranges at $\pm 2\sigma$ in Figure 14. This demonstrates the consistent performance of our method. All main experiments presented use the data loader with random seed 42.

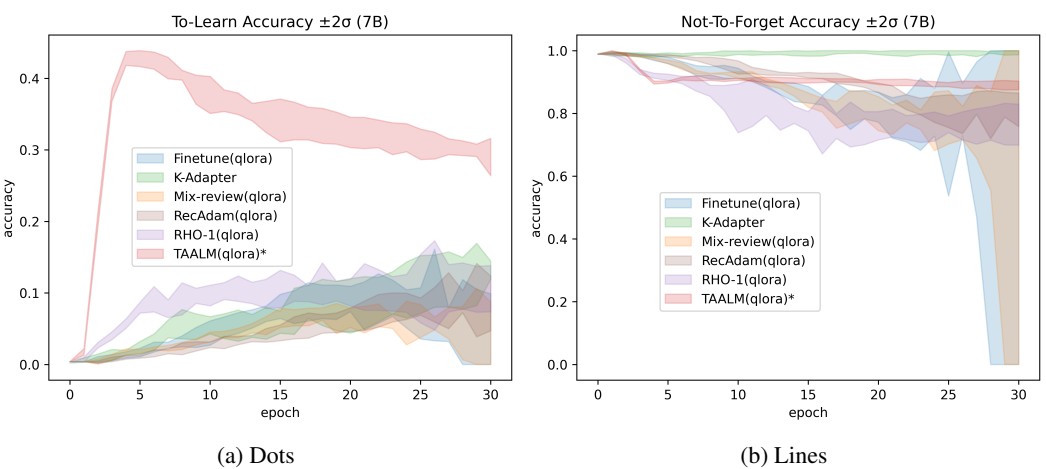

(a) Dots

(b) Lines

Figure 14: Performance of the large-scale baseline models (Llama2-7B) on LAMA-CKL, depicted with $\pm 2\sigma$ error ranges. Results are calculated from five random trials, employing different random seeds over the train data-loader.

# B  Descriptive Form & Schematic Form

Each unit of the LAMA dataset comprises a knowledge base triple and a corresponding sentence that encapsulates the information contained within the triple. These sentences are presented in a short *descriptive form.* In contrast, our newly proposed **schematic form** organizes these triples more systematically. The template and example are described in Figure 15. We propose the schematic form because it better aligns with the uni-directional nature of causal language models (LMs). In the descriptive form, critical information often comes after the label tokens, which can be problematic. For instance, in the descriptive form template "[X] is [Y] citizen", the causal LM lacks the crucial cue 'citizen' when predicting [Y]. This not only makes the assessment less accurate but also introduces noise into the train-attention learning process.

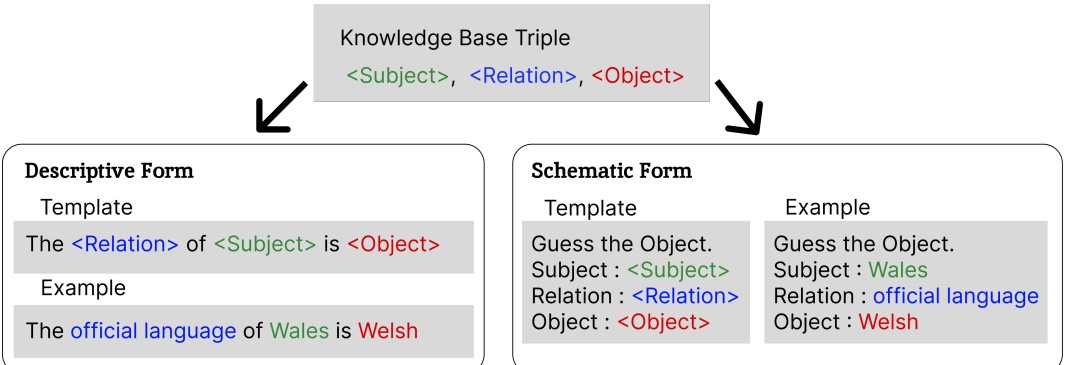

Figure 15: Descriptive form and our proposed schematic form template to rearrange knowledge triple into a human readable document.

# C  Detail of TEMPORALWIKI Experiment

## C.1  Corpus setup

The original corpus consists of two datasets to learn for 4 periods (0809, 0910, 1011, 1112); (1) Wikipedia: the whole snapshot of Wikipedia documents of each period. (2) DIFFSETs: the segments of documents that only contain changed information compared to the last period snapshot. The corpus also includes test datasets (TWiki-Probes) corresponding to each period. Just like the LAMA-CKL, the Wikipedia snapshots and DIFFSETs consist of evidence documents, while TWiki-Probes consist of knowledge base triples. Because we used the data from the very first period to train Train-Attention, only the data from the next three periods (0910, 1011, 1112) are used for the evaluation.

The original study shows that learning only the DIFFSET results in better performance than learning the entire snapshot. So we excluded the baselines that learn full snapshots and set learning of DIFFSET as the default. We also employed heuristic filtering for the DIFFSET. Empty texts containing only 'nan' in the content, or texts with more than 70% of non-letters, which are assumed to contain little meaningful information, are filtered out. Therefore the total number per DIFFSET data decreases from an average of 837K to 707K documents per each dataset.

Each Twiki-Probes consists of two parts: (1) CHANGED: assumed to consist only of knowledge base triples that contain changed information compared to the last period. (2) **Unchanged**: contain only retained information from the last period, which is the complementary set of the CHANGED. CHANGED set is supposed to measure plasticity, while UNCHANGED set is supposed to measure stability.

## C.2  Evaluation and baseline setup

We follow the evaluation settings of the original work Jang et al. [2022]: a model continually learns train datasets of each period while evaluating performance on TWiki-Probes after training of each period. The evaluation metric is the mean perplexity of object label words. One dataset is updated for only 1 epoch.

The original study simply arranges the triples in order (e.g., "Wales official language Welsh") and sets the task as predicting the object label token. As this form of sentence might seriously violate the grammar, it has been reported that perplexity tends to pick extraordinarily high in the original work. To handle this, we adopt the schematic task format used in LAMA-CKL (§4.1).

Baselines and hyper-parameters settings follow that of LAMA-CKL (§4.1), which follow the Jang et al. [2022].

## C.3 Train-Attention training setup

The DIFFSET and Twiki-Prob of the first period (0809) are used for training Train-Attention. To train Train-Attention (Algorithm 1), data ($\mathcal{D}$) and the corresponding task ($\mathcal{T}_\mathcal{D}$) must be paired, while the documents in DIFFSET and knowledge base triples in Twiki-Probes are not paired. To tackle this challenge, we search the DIFFSET documents which contain both the object and subject of the Twiki-Probes unit and manually join them. The total number of pairs is 5235. Validation setting and hyper-parameters follow §4.1.

## C.4 Analysis on the results

In this section, we present the detailed experimental result on TEMPORALWIKI, specifically involving the discussion in §4.3, on the reason we propose the new benchmark LAMA-CKL. Figure 16 indicates that performances in both Changed and Unchanged mostly move in the same direction when the baselines are trained on the DIFFSET of each period, presenting a rare trade-off between plasticity (Changed) and stability (Unchanged). This is contrasting to the observation on our new benchmark, LAMA-CKL (Figure 13), which exhibits a clear trade-off between plasticity (TO-LEARN) and stability (NOT-TO-FORGET). This suggests that the LAMA-CKL is a better benchmark for observing the trade-off between plasticity and stability.

Table 7 presents the portion of task sentences in each Changed and Unchanged set, which have evidence documents in each period of the training dataset (DIFFSET). An average of 8.0% of every Unchanged set have evidence documents in the DIFFSET of the corresponding period, indicating that training on DIFFSET partially results in an improvement in the performance on the UNCHANGED set. The Unchanged set from the last period (TWiki-Probes-1112) contains evidence documents in the DIFFSETs from all three preceding periods, suggesting that its supportive effect is cumulative. This is a complication factor for observing stability.

Table 7: The portion of units in each period of TWiki-Probes, which have corresponding evidence document in the DIFFFSET. **Un** and **C** indicate Unchanged and Changed set, respectively.

|  | TWiki-Probes-0910 | | TWiki-Probes-1011 | | TWiki-Probes-1112 | |
|---|---|---|---|---|---|---|
|  | **Un** | **C** | **Un** | **C** | **Un** | **C** |
| DIFFSET-0910 | 7.8% | 83.6% | 7.6% | 39.5% | 7.7% | 25.3% |
| DIFFSET-1011 | - | - | 7.4% | 81.8% | 7.9% | 26.2% |
| DIFFSET-1112 | - | - | - | - | 8.7% | 83.3% |

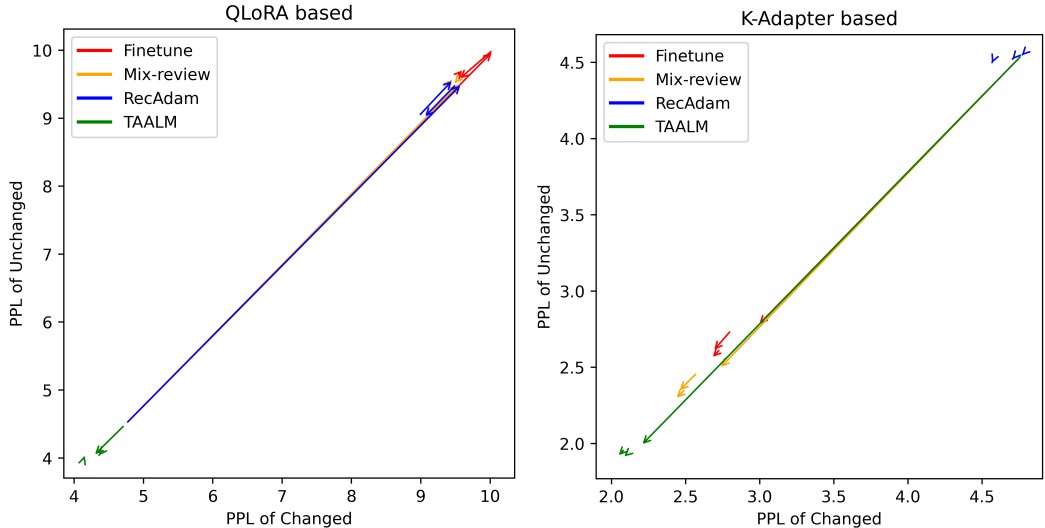

Figure 16: The experimental result of small(TinyLlama-1B) baselines on TEMPORALWIKI benchmark. The x-axis and y-axis correspond to the perplexity of Changed and Unchanged sets, respectively. Each color represents baselines, and the arrows represent the experimental result from each period. The start point of the arrow indicates the performance before training, while the endpoint indicates the performance after training.

## D    Reduction of resources through TA on BERT

Due to the substantial GPU resources required to train the TA, it is necessary to find ways to reduce resource consumption. A promising approach is utilizing Bidirectional Transformer (BERT) as a body for TA, which has high inferential capabilities even at a very small size (108M) compared to the previous body (Tinyllama 1.1B), due to its bidirectional property. Since BERT has a different tokenizer from our generation model, the Llama family, we integrate BERT with the Llama2 tokenizer and pre-train it for one epoch on 17GB Wikipedia documents (9 days using 8 of 24GB GPUs). Then, we finetune this BERT as TA, paired with the generation model of 1B (Tinyllama). This very lightweight TAALM is sufficiently trained on only a single 24GB GPU, significantly reducing resource use compared to the previous version (single 82GB GPU), thus making it affordable for the general environment. On the inference, the TA on BERT demonstrates compatibility with both the 1B and 7B generation models. Although its performance is below that of the TA on Llama, it still exhibits the highest performance among the other baselines.

Table 8: LAMA-CKL performance of small (TinyLlama-1B) baselines.

(a) Baselines with large generation model (Llama2 7B)

|            | Parameter size of TA | Top Acc | Epoch | NF Acc | Total Knowledge |
|------------|---------------------|---------|-------|--------|-----------------|
| Finetune   | NA                  | 0.1150  | 16    | 0.8174 | 0.9324          |
| TA (Llama) | 1.1B                | **0.4290** | **4** | 0.8983 | **1.3273**      |
| **TA (BERT)** | 108M             | 0.3210  | 6     | **0.9388** | 1.2598       |

(b) Baselines with small generation model (Tinyllama 1B)

|            | Parameter size of TA | Top Acc | Epoch | NF Acc | Total Knowledge |
|------------|---------------------|---------|-------|--------|-----------------|
| Finetune   | NA                  | 0.0700  | 29    | 0.7693 | 0.8393          |
| TA (Llama) | 1.1B                | 0.3260  | **4** | 0.9078 | **1.2338**      |
| **TA (BERT)** | 108M             | **0.2440** | 9  | **0.9267** | 1.1707       |

# E  Ablation

Based on the token importance predicted by the TA, various design choices are possible. We explore and compare the effectiveness of these variations. This study enhances our understanding of how generative LM interacts with token weights when learning data. The description of the components and experimental results are as follows.

**(1) Token-importance weight (ours) :** The original variation that utilizes token-importance weight predicted by TA for target-weighted learning. **(2) Known token masking :** Masking out the tokens in real-time when prediction and label matches. This method is intended to enhance "model awareness" in TA, as TA is more oriented to "task awareness." **(3) Token weight dropping :** Among token-weight generated by TA, dropping weights that are below the top k% levels. We tested 50% and 80%. Vanilla TA is the same as the threshold of 0%. This method is intended to cut out noisy targets, as TA is supposed to assign lower weight to un-useful tokens.

**Results**    Known token masking does not yield better results compared to TAALM w/ token-importance weight. We hypothesize that the effect of known masking is limited because task awareness is already achieved when the loss of learned tokens is reduced. Test results on the TAALM w/ token weight dropping show that as the threshold increases, the top accuracy decreases. This suggests that some useful targets are mixed in among the lower weights, and it helps the model learn better somehow. On the contrary, not-to-forget accuracy slightly improves as the threshold increases. This seems as the effect of 1) cutting out noisy targets and 2) trade-off for lower learning. However, Total Knowledge is best on the TAALM w/ token-importance weight (ours). Overall experimental results indicate that, since TA is optimized to maximize task performance, adding heuristic interventions appears to produce suboptimal outcomes.

Table 9: LAMA-CKL performance of Llama2-7B based baselines.

|  | Top Acc | Epoch | NF Acc | Total Knowledge |
|---|---|---|---|---|
| Finetune | 0.1150 | 16 | 0.8174 | 0.9324 |
| TAALM w/ token-importance weight (**ours**) | **0.4290** | **4** | 0.8983 | **1.3273** |
| TAALM w/ known token masking | 0.3920 | **4** | 0.9075 | 1.2995 |
| TAALM w/ token weight dropping < 0.5 | 0.4100 | 7 | 0.9148 | 1.3248 |
| TAALM w/ token weight dropping < 0.8 | 0.3850 | **4** | **0.9267** | 1.3117 |

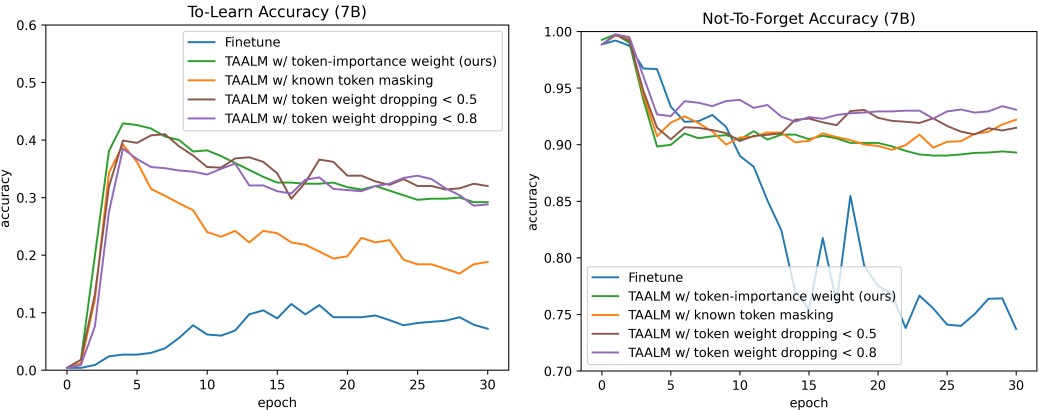

Figure 17: Ablation study on various design choices with token importance from Train-Attention (TA). TA is Tinyllama-1B, and the generation model is Llama2-7B.

# F  Analysis on the attention pattern of TA

TA is observed to generally assign attention to proper nouns, nouns, and verbs that contain the subject's character. The focus of attention seems to be diverse depending on the content of the text. For autobiographical texts, TA shows a tendency to focus on words that represent the person's occupation or major events (Figure 18a). In passages listing regional relations, TA pinpoints the names of locations (Figure 18b). This appears to be due to the consideration of probable queries. While TA (trained on LAMA-CKL) omits some words in the documents, it tends to not miss location names. This is likely because many queries in the LAMA-CKL benchmark involve location-related aspects (e.g., birthplaces, location of the workplace).

We also provide an attention map of TA trained on the multi-session chat (Figure 18c). Here, we regard prior dialogue sessions as data ($\mathcal{D}$) and an understanding of the next session as task ($\mathcal{T}_{\mathcal{D}}$). Unlike Wikipedia documents, chit-chat dialogues contain fewer useful words, highlighting the necessity for TA. TA focuses on the interlocutor's information like the occupation and pet's name.

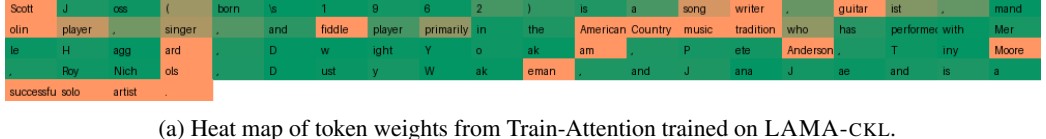

(a) Heat map of token weights from Train-Attention trained on LAMA-CKL.

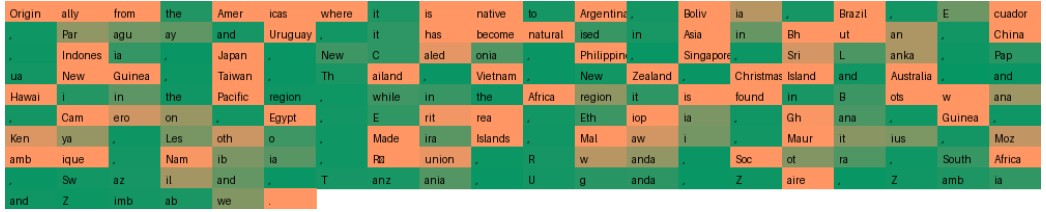

(b) Heat map of token weights from Train-Attention trained on LAMA-CKL.

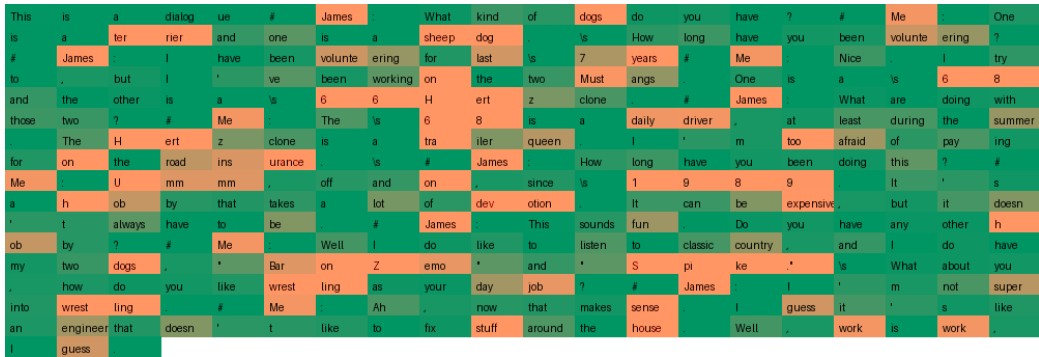

(c) Heat map of token weights from Train-Attention trained on Multi-Session Chat dataset.

Figure 18: Heat map of token weights from Train-Attention. Orange color indicates higher weights.

