# OpenReview forum: "Train-Attention: Meta-Learning Where to Focus in Continual Knowledge Learning"
_NeurIPS.cc/2024/Conference — NeurIPS 2024 poster_

### Official Review · Reviewer_dYuZ · 2024-06-27

**Soundness:** 3
**Presentation:** 3
**Contribution:** 3
**Rating:** 6
**Confidence:** 3

**Summary:**

This paper proposes a meta-learning framework to dynamically adjust token perplexity weights based on their usefulness to achieve good continuous knowledge learning performance and also provides a new benchmark for CKL.

**Strengths:**

1. It is inspiring to use meta-learning technique for adjusting token weight.

2. The proposed benchmark improves on distinguishing plasticity and stability.

3. The experiments are comprehensive and solid.

**Weaknesses:**

1. The idea of adjusting token weights is not fresh enough and shares some similarity with [1]. However, training a meta-learner to evaluate token importance is still a good try.

2. The analysis about token importance (Figure 6 and its corresponding analysis) is not enough. Analysis about attention pattern on these tokens would be a good supplement.


[1] Lin Z, Gou Z, Gong Y, et al. Rho-1: Not all tokens are what you need[J]. arXiv preprint arXiv:2404.07965, 2024.

**Questions:**

1. In Figure 7, I am wondering why the to-learn accuracy drops after 4-th epoch? It is kind of counter-intuitive to me as I expect it will grow along with increasing epoch.

2. Could the authors provide more analysis and rationale behind the phenomena that K-adapter keeps its not-to-forget accuracy basically unchanged?

**Limitations:**

The authors addressed the limitations adequately.

---

> ### Author Rebuttal · Authors · 2024-08-06
>
> **[W1] The idea of adjusting token weights is not fresh enough and shares some similarity with “RHO-1”. However, training a meta-learner to evaluate token importance is still a good try.**
>
> We truly appreciate your recognition of our effort, as well as your deep understanding of our paper.
>
> As you mentioned, in addition to the concurrent work RHO-1, earlier studies such as token dropping [1] have also addressed methods for adjusting token weights, and these works are highly inspiring. However, we introduced our work’s novelty from these methods in Section 2, and the performance comparison with the RHO-1 in Section 4.
>
> For our best understanding, RHO-1 assumes the loss difference between the current training model and the reference model as a measure of importance. It indicates that lower confidence correlates with higher importance. This shares similar concept with previous work of token dropping [1] and focal loss [2].
>
> However, these methods still allow many tokens to be learned that a sufficiently trained mature model does not need, mainly due to its naive definition of “token importance”. Therefore, these techniques have been introduced more as pretraining methods rather than as CKL approaches.
>
> In contrast, we define token importance as "usefulness" rather than confidence, and propose a perspective that the acquisition of knowledge should be optimized for the task. This novel perspective enables meta-learning methods to be applied to token weighting. This is the key distinction between our method and previous token weighting approaches.
>
> Our work's effectiveness is also demonstrated through experiments. The result in Table1 shows that our approach is significantly more optimal than RHO-1 for CKL scenario.
> ***
> **[W2] The analysis about token importance (Figure 6 and its corresponding analysis) is not enough. Analysis about attention pattern on these tokens would be a good supplement.**
>
> For your suggestion, we include heat-maps of token importance from TA, in the **rebuttal pdf**.
>
> TA is observed to generally assign attention to proper nouns, nouns, and verbs that contain the subject's character. The focus of attention seems to be diverse depending on the content of the text. For autobiographical texts, TA shows a tendency to focus on words that represent the person's occupation or major events (Figure 1). In passages listing regional relations, TA pinpoints the names of locations (Figure 2). This appears to be due to the consideration of probable queries. While TA (trained on LAMA-ckl) omits some words in the documents, it tends to not miss location names. This is likely because many queries in the LAMA-ckl benchmark involve location-related aspects (e.g., birthplaces, location of the workplace).
>
> We also provide an attention map of TA trained on the multi-session chat [3] (Figure3). Here, we regard prior dialogue sessions as data (D) and an understanding of the next session as task (T_D). Unlike Wikipedia documents, chit-chat dialogues contain fewer useful words, highlighting the necessity for TA. TA focuses on the interlocutor’s information like the occupation and pet's name.
> ***
> **[Q1] In Figure 7, I wonder why the to-learn accuracy drops after the fourth epoch. It is kind of counterintuitive to me, as I expect it to grow along with the increasing epoch.**
>
> As the reviewer observed, in the main experiment (Figure 7), the performance declines after a rapid peak.
>
> We hypothesize this is a result of overfitting. While the token weights from TA include beneficial targets, there must also be some that are not. Performance achieves peak until the model completes learning for the true target, but learning may continue for the false targets. This continued learning leads to parameter updates in suboptimal directions, resulting in forgetting.
>
> The comparison between TA and the oracle (Figure 8) provides evidence for this rationale. As the oracle's performance does not decline with further training, the differences in accuracy trends could be caused by false targets.
>
> This phenomenon can also be interpreted as a type of overfitting where the model becomes too fitted to the training data which has different distribution from test data. As the phenomenon of declining after peaking is common due to overfitting, this cycle seems to occur more rapidly for TAALM.
> ***
> **[Q2] Could the authors provide more analysis and rationale behind the phenomena that K-adapter keeps its not-to-forget accuracy basically unchanged?**
>
> As depicted in Figure 7, K-Adapter shows a slower increase in to-learn accuracy and a slower decrease in not-to-forget accuracy compared to baselines using QLoRA adapter. This phenomenon appears to be due to structural differences between K-Adapter and QLoRA.
>
> QLoRA, a quantized version of Lora, adds a new parameter, C (the result of multiplying two smaller matrices, A and B), to the original model parameters. As the original parameters can't be recovered after C is added, Lora is essentially the same as modifying the original parameters. Modifying parameters means that the model forgets other knowledge that is previously held by the parameters.
>
> In contrast, the K-Adapter is a separate transformer layer that takes the hidden states of the original model as inputs. This layer includes features like gates and residual networks, which can choose to let the original outputs pass through without changing them.
>
> To summarize, while QLoRA modifies the existing model parameters and stores only the changes in the adapter, while K-Adapter does not directly alter the parameters but instead trains an additional independent transformer alongside them. This approach makes K-Adapter slower in learning, but it seems to retain more of the original knowledge.
> ***
> _**References**_
> [1] Token dropping for efficient bert pretraining
> [2] Focal loss for dense object detection
> [3] Beyond Goldfish Memory: Long-Term Open-Domain Conversation

---

> > ### Comment · Reviewer_dYuZ · 2024-08-11
> > **Respond**
> >
> > Many thanks to the author for the responses. They address my concerns and I will maintain a positive score.

---

> > > ### Author Response · Authors · 2024-08-12
> > > **Sincere thanks**
> > >
> > > We are glad that our reply addressed your concerns. Your feedback has strengthened our work. We will incorporate your feedback into the camera-ready version. We would be happy to address any further questions/suggestions that might come up until the end of the discussion period.

---

### Official Review · Reviewer_Az1R · 2024-07-07

**Soundness:** 3
**Presentation:** 2
**Contribution:** 3
**Rating:** 6
**Confidence:** 4

**Summary:**

This paper studies the continual knowledge learning (CKL) problem in large language models. The authors notice that the existing methods either apply equal weights to all tokens or re-weight tokens with a trivial consideration that tokens with low confidence are important. They propose a new definition of token importance based on the expected functionality of the token in solving related tasks. In particular, they design a meta-learning strategy to learn token weights in the training phase to facilitate the continual learning of new knowledge while retaining already learned knowledge. A new benchmark, LAMA-CKL, is established. The experimental results demonstrate that the proposed method outperforms other methods in both existing and introduced benchmarks.

**Strengths:**

1. Consider the expected functionality of tokens in solving related tasks as token importance is novel and well-motivated.
2. The proposed meta-learning framework is simple yet effective and can be complementary to other methods.
3. A new benchmark for evaluating continual knowledge learning is established.

**Weaknesses:**

1. Naming token importance as "usefulness" is too broad and does not accurately reflect the motivation of the paper.
2. Meta learning is a well-known method, the description of the algorithm details in Section 3.2 is however confusing and unclear.
3. The proposed method seems to only be applicable to the experimental setup described in the paper. This setup requires pairs of \mathcal{D} and \mathcal{T}_\mathcal{D}, where \mathcal{T}_\mathcal{D} represents a task that can be solved using the information contained in \mathcal{D}. This setup is not general enough and is limited to very specific scenarios.

**Questions:**

Why does the proposed meta learning approach reach performance peaks in fewer epochs compared to other methods? Is there an explanation for this?

**Limitations:**

The discussion of limitations is included in the paper.

---

> ### Author Rebuttal · Authors · 2024-08-06
>
> **[W1] Naming token importance as "usefulness" is too broad and does not accurately reflect the motivation of the paper.**
>
> Thank you for the constructive feedback on the clarity of expression. We consider  "task-utility" as it more clearly implies usefulness in relation to the task.
> ***
> **[W2] Meta learning is a well-known method, the description of the algorithm details in Section 3.2 is however confusing and unclear.**
>
> For your feedback, we briefly restate using the frame of legacy meta-learning work such as MAML [1], meta-SGD [2] as follows:
>
> Meta-learning consists of an inner loop that trains the model and an outer loop that trains the meta-learner. The true goal is to train the meta-learner. In our work, the meta-learner is called Train-Attention (denoted ϕ), and the model is a generative LLM (denoted θ).
>
> **Inner loop**: θ learns from data (D), updated into θ′. When learning, θ utilizes the token importance (W_ϕ), which is predicted by ϕ, in a target-weighted manner. See Eq.(4)
>
> **Outer loop**: ϕ learn to generate W_ϕ to maximize task (T_D) performance of θ′. See Eq.(5)
>
> Each time the outer loop is executed, ϕ is updated, and θ is repeatedly reset to the initial point.
>
> Compared to [1][2], the meta-learner in MAML is the initial parameter of that model, and the meta-learner in meta-SGD is a hyperparameter (e.g., learning rate). In contrast, our meta-learner is a separate model that predicts the hyperparameter.
>
> In the camera-ready version, we will adopt this frame to explain more clearly.
> ***
> **[W3] The proposed method seems to only be applicable to the experimental setup described in the paper. This setup requires pairs of D and T_D, where T_D represents a task that can be solved using the information contained in D. This setup is not general enough and is limited to very specific scenarios**
>
> We appreciate the reviewer for highlighting this fundamental question. We assume that our method is applicable on the scenario where LM should continually update its knowledge. And a pair of D and T_D represent quite a general situation, more than it looks.
>
> It is easy to think that information exists in isolation from practical use, but its value of existence comes up when it finds somewhere to be used. In other words, every data (D) potentially has its task (T_D), and how to find them and link them as a pair is up to future research. We introduce practical examples below.
>
> **1)The common retrieval augmented QA scenario is applicable**
>
> Any retrieval augmented question answering (QA) scenarios are suitable for applying TAALM. Currently, the main solution for this scenario is to retrieve the related document and append it to the prompt while generating answers. However, if we regard the retrieved document as D and the question-answer pair as T_D, it will build an TAALM that is capable of continual knowledge updates through retrieving. Since most current uses of LMs fall into this retrieval-augmented QA scenario, we believe our methodology has significant applicability.
>
> **2)Multi-session dialogue scenario is also applicable**
>
> Furthermore, depending on how we interpret D and T_D, the application of TAALM can become even broader and more general. Consider a scenario involving multiple conversation sessions with our friends. For instance, if a friend says, "I'm moving to Washington", in the next meeting, we should remember this move to have a good conversation. In this case, the earlier conversation can be interpreted as D, and the utterance generation task of the subsequent conversation as T_D.
>
> In practice, we successfully trained TAALM using this approach on the Multi-Session-Chat (MSC) [3] dataset, treating one session as D and the next as T_D. This training approach enhances the model’s dialogue understanding performance in the subsequent sessions compared to standard fine-tuning, which we plan to describe in the following research. We provide a token importance heat map, which is generated by the TA fitted to the MSC dialogue (Figure 3 in the rebuttal pdf).  And we will introduce this approach in the camera-ready version.
>
> **[Q1] Why does the proposed meta learning approach reach performance peaks in fewer epochs compared to other methods? Is there an explanation for this?**
>
> We truly appreciate the reviewer's careful observation. We provide an easy and detailed version of explanations here.
>
> **Easy explanation**
>
> As the TA is optimized to predict token weights that maximize the task performance of LM, the LM reaches the performance peak in fewer epochs with the optimal token weights.
>
> **Detailed explanation**
>
> We hypothesize that the interference among the gradient vectors occurs in the standard finetuning, resulting in slower learning.
>
> For instance, assume there is a sentence to learn, and five chunks of information (denote A, B, C, D, E) in that one sentence. Among the five, only A is useful and worth learning, while others are not. Assuming the model size as 4-dimension for simplicity, the gradient vector (necessary parameter change for the model to learn each piece of information) for each are: A as [1,1,0,-1], B as [1,1,0,1], and C as [-1,-1,1,0],  et cetera. In this case, the directions of the first and second elements for A and C cancel each other, and the last element of A and B cancels. These interferences will delay the parameter update to the necessary amount to reach A.
>
> As seen in this case, the more diverse the information to be learned, the more probable that gradient vectors will cancel each other out. This results in extended training steps required to learn the targeted information A. In contrast, TAALM makes the model selectively learn only targeted information, A, thus avoiding interference from other data and enabling faster learning.
> ***
> _**References**_
> [1] Model-agnostic meta-learning for fast adaptation of deep networks
> [2] Meta-sgd: Learning to learn quickly for few-shot learning
> [3] Beyond Goldfish Memory: Long-Term Open-Domain Conversation

---

> > ### Author Response · Authors · 2024-08-12
> > **Sincere thanks**
> >
> > We thank again for your constructive review. We hope that our responses have adequately addressed your previous concerns, and we would be happy to address any further questions/suggestions that might come up until the end of the discussion period.

---

### Official Review · Reviewer_6R4i · 2024-07-10

**Soundness:** 3
**Presentation:** 3
**Contribution:** 2
**Rating:** 5
**Confidence:** 4

**Summary:**

The article introduces Train-Attention Augmented Language Model (TAALM), a novel approach for continual knowledge learning in large language models. TAALM dynamically assigns weights to tokens based on their usefulness, optimizing learning efficiency and minimizing forgetting.

**Strengths:**

- This paper introduces Train-Attention Augmented Language Model (TAALM) to reweight token sequences in tuning LLM to avoid catastrophic forgetting.
- It introduces a new benchmark, LAMA-CKL, for clearer learning-retention trade-off assessment.
- TAALM demonstrates state-of-the-art performance, compatibility with existing methods, and computational efficiency, advancing the field of CKL.

**Weaknesses:**

- The proposed method is not novel and it is widely used in multi-task learning to adjust task loss weights in the learning process, like "MetaWeighting: Learning to Weight Tasks in Multi-Task Learning".
- This paper lacks of explanation of why it could maintain previous knowledge without forgetting.

**Questions:**

- What will be the influence of dropping the lightweight token sequences in training? If only important token sequences matter in learning new knowledge.
- Can authors offer more theoretical or experimental explanations for the effectiveness of the trained attention?
- I was confused about the relation of the learned meta weight with continual learning. Why dropping the unuseful tokens will help the model learn without forgetting?
- The ablation study of the learned meta-weight is missing. I only found the results in Figure 6, however the effectiveness of the learned weight should be further validated.

---

> ### Author Rebuttal · Authors · 2024-08-06
>
> First of all, we appreciate your constructive feedback on our work. However, it seems there might be some misunderstandings about our work. Most of the queries raised are addressed in Section 2; we recommend reexamining this section for clarification. Before addressing the issues raised by the reviewer, we would like to provide a summary of our work.
>
> **1)Mature LM learns not only necessary tokens**
>
> Our research aims to address the inefficiencies of the traditional fine-tuning procedure of LMs, thus enhancing its continual knowledge learning (CKL) capacity. What are the inefficiencies in the fine-tuning procedure? Even though the model only needs a part of the information in a sentence, it has to learn all the tokens in the whole sentence.
>
> As shown in Figure 1 of our paper, let's assume a language model (LM) learns the sentence "the president of the US is Biden". If this LM were a scratch model that hasn't learned anything yet, it would need to learn the information in every token sequence, as all sequences contain at least information on grammar rules. However, if this model were a mature model already pretrained and finetuned, it would not need all the information. If this model were trained in 2020 and incorrectly believes that the current president is Trump, then in this sentence, the only token that carries important information would be "Biden."
>
> **2)Restrict unnecessary parameter changes to avoid forgetting**
>
> What if we force this mature model to still learn every token in the sentence? Learning means adjusting parameters, and adjusting parameters means forgetting other knowledge which is contained in those parameters. On the contrary, what if we limit the learning scope to only the essential parts and restrict the model from learning the rest? By minimizing parameter adjusting, the model could keep its other knowledge from forgetting. (**W2, Q3**)
>
> **3)Train-Attention (TA) decides what is necessary**
>
> The role of token importance (meta-weight) is to determine which tokens are important. In the example above, a high weight would be assigned to "Biden," and no weight would be assigned to the other tokens. To determine the “importance” of each token, we propose “usefulness” as a new criterion. Just as a person might remember the name "Biden" to avoid being foolish in future conversations, a model learns specific knowledge only to perform well in future tasks. Therefore, we use meta-learning to identify which tokens contain information that is useful for future tasks. (**Q2, Q3**)
>
> We have demonstrated this sufficiently through experiments. First, through an experiment with the oracle label (Figure 8), we showed that focusing solely on necessary information improves learning speed, capacity, and retention. Additionally, our main experiment (Figure 7, Table 1) proved that the weights predicted by our TA are well-predicted and perform similarly to the oracle. (**Q2**)
> ***
> **[W1] The proposed method is not novel and it is widely used in multi-task learning to adjust task loss weights in the learning process, like "MetaWeighting ".**
>
> Thank you for recommending an excellent paper. However, it is essentially different from our research.
>
> First, MetaWeighting is a method of assigning weights to each task when conducting multi-task learning. This work can be categorized as “hard mining” (e.g., hard example mining, hard task mining), and the motivation of this work is to refine and automate this hard mining process using meta-learning. Methodology: MetaWeighting adjusts the learning rate of each task, and within the same task, all data are learned with a uniform weight.
>
> On the other hand, our work is intended to improve the model’s CKL capacity by addressing the inefficiencies in the standard finetuning procedure, which is caused by uniform weighting through all tokens. Our contributions are 1) The observation that uniform weighting is particularly detrimental to CKL, 2) suggesting a novel view that redefines token importance as usefulness, and 3) this view enables meta-learning to be applied to this problem. These are the novelty of our work and are not similar to MetaWeighting. Methodology: Weights are applied to the tokens of the data being learned, and are optimized to enhance the performance of individual tasks. This is different from meta-weight where weights are applied to the task itself.
>
> As described, our work and meta-weight research only share the keywords "task" and "meta-learning". Beyond these, the motivation, implementation pipeline, and objectives are all different.
>
> ***
>
> **[W2] This paper lacks of explanation of why it could maintain previous knowledge without forgetting. [Q2] Can authors offer more theoretical or experimental explanations for the effectiveness of the trained attention? [Q3] I was confused about the relation of the learned meta weight with continual learning. Why dropping the unuseful tokens will help the model learn without forgetting?**
>
> We address these issues in the prior part of our response. And each is also explained in our paper, mainly on Section 2.
>
> ***
> **[Q1] What will be the influence of dropping the lightweight token sequences in training? If only important token sequences matter in learning new knowledge.  [Q4] The ablation study of the learned meta-weight is missing. I only found the results in Figure 6, however the effectiveness of the learned weight should be further validated.**
>
> We really appreciate you for the insightful suggestion. Investigating this issue has led us to a deeper understanding of our own work.
> Based on the token importance predicted by the TA, various design choices are possible, including your suggestion. We explore and compare the effectiveness of these variations. This study enhances understanding of how generative LM interacts with token weights when learning data. Detailed descriptions of the components and experimental results can be found in the **global rebuttal**.

---

> > ### Comment · Reviewer_6R4i · 2024-08-11
> >
> > Many thanks to the authors for the detailed answers. After considering both the other reviews and the rebuttals, I will increase my score.

---

> > > ### Author Response · Authors · 2024-08-12
> > > **Sincere thanks**
> > >
> > > Thank you for recognizing our work and for raising your score. Your feedback has strengthened our work. We would be happy to address any further questions/suggestions that might come up until the end of the discussion period.

---

> > > ### Comment · Area_Chair_Xf9o · 2024-08-13
> > > **Re: Official Comment by Reviewer 6R4i**
> > >
> > > Dear Reviewer 6R4i,
> > >
> > > Having gone through the paper and the discussion here, it appears that the general consensus is leaning towards acceptance. If the authors have addressed your concerns. Please adjust your scores accordingly.
> > >
> > > Regards,
> > >
> > > AC

---

### Official Review · Reviewer_CkYa · 2024-07-20

**Soundness:** 3
**Presentation:** 3
**Contribution:** 2
**Rating:** 6
**Confidence:** 4

**Summary:**

The paper introduces Train-Attention-Augmented Language Model (TAALM), a novel approach to continual knowledge learning (CKL) in large language models (LLMs). Unlike traditional methods that uniformly apply weight across all tokens, TAALM dynamically predicts and applies weights to tokens based on their importance using a meta-learning framework. This approach aims to enhance learning efficiency by targeting essential knowledge updates and minimizing forgetting. The authors also introduce a new benchmark, LAMA-CKL, to better assess the trade-off between learning new information and retaining existing knowledge. Experimental results show that TAALM significantly outperforms existing CKL methods on both new and established benchmarks.

**Strengths:**

Originality: The paper introduces a novel approach to CKL by using meta-learning to predict token importance, which is a significant departure from traditional methods.
Quality: The technical claims are well-supported by both theoretical justifications and empirical evidence. The experiments are comprehensive and rigorously conducted.
Clarity: The paper is well-structured and clearly written, with detailed explanations of the methodology and results.
Significance: The contributions are substantial, offering both a new method (TAALM) and a new benchmark (LAMA-CKL) that advance the state-of-the-art in CKL research.

**Weaknesses:**

Generalization to Other Tasks: While the paper demonstrates the effectiveness of TAALM in the context of CKL, it would be beneficial to explore its applicability to other types of continual learning tasks beyond language models.
Computational Resources: The training of TAALM, especially with large models, requires significant computational resources. A discussion on the scalability and efficiency of the approach for smaller models or resource-constrained environments would be useful.
Ablation Studies: While the paper includes comprehensive experiments, additional ablation studies to isolate the impact of different components of TAALM (e.g., the specific meta-learning algorithm used) would strengthen the evaluation.

**Questions:**

Can the authors provide more details on the potential applicability of TAALM to other types of continual learning tasks outside of language models?
How does TAALM perform with significantly smaller models or in environments with limited computational resources?
Could the authors include more detailed ablation studies to better understand the contribution of each component of TAALM to the overall performance?

**Limitations:**

The paper adequately discusses the limitations of the proposed approach, particularly the task-specific nature of Train-Attention and the requirement for data-task pairs for training. The authors also highlight the potential for Train-Attention to evolve and adapt to new tasks, suggesting areas for future exploration. However, a more detailed discussion on the computational requirements and scalability of TAALM would be beneficial.

---

> ### Author Rebuttal · Authors · 2024-08-06
>
> We truly appreciate your deep understanding of our paper. We address the key issue raised by the reviewer in the comments below:
> ***
> **[W1+Q1] Can the authors provide more details on the potential applicability of TAALM to other types of continual learning tasks outside of language models?**
>
> Thank you for the constructive feedback. The applicability of TAALM to others is also our main concern. However, we first wish to explain that continual "knowledge" learning (CKL) is a critically important issue, while it has been relatively understudied.
>
> **1)TA handles the unique problem of LM**
>
> The CKL problem is mainly related to LM because possession of knowledge is a unique characteristic of LM. This issue is distinct from traditional continual learning (CL), which primarily focuses on learning new "tasks" rather than new "knowledge". When we ask well-known large LMs (e.g., chatGPT) "Who is the current president?", most reply that, they cannot answer as their knowledge was updated last in 2023 or any previous years. This issue arises because the models lack the capability for CKL.
>
> LM is also different from other domains as it learns sequences. This property yields unique inefficiencies, resulting in extra forgetting. TAALM is specifically designed to tackle this problem. Despite the importance and uniqueness of this CKL issue, research has been conducted mainly by importing general CL approaches from fields outside of LM. That is why this specified study is valuable.
>
> **2)Applicable for other sequence learning: reinforcement learning**
>
> Since our methodology is specialized for sequence learning, we expect it to be applicable for reinforcement learning (RL), which also involves learning of sequence (i.e., Markov Decision Process). Specifically, it seems suitable for addressing the credit assigning problem of sparse reward tasks.
>
> In the field of RL, one of the major challenges is to learn tasks with sparse rewards. Sparse reward tasks are those where rewards are given only at the end of very long trajectories. Traditional methods like temporal difference learning have a problem with the short observation window, which is considered not suitable to handle this problem. [1]
>
> The common solution is imitation learning, which is memorizing an entire successful demonstration trajectory without considering rewards. However, it forces a model to absorb all inefficient movements included in the demonstration without self-refining. To solve this, it is necessary to elucidate the impact of each action step (i.e., credit assignment). [2]
>
> Our Train-Attention (TA) is suitable for addressing this credit assignment. TA predicts the importance of each token based on its impact on the final performance, which is very similar to predicting the importance of each action step of a long trajectory.
> ***
> **[W2+Q2] How does TAALM perform with significantly smaller models or in environments with limited computational resources?**
>
> Thank you for the insightful question. This issue is a primary concern for us, and we aim to address it in the subsequent study.
>
> A promising approach is utilizing a Bidirectional Transformer (BERT) as a body for TA, which has high inferential capabilities even at a very small size (108M) compared to the previous body (Tinyllama 1.1B), due to its bidirectional property.
>
> Since BERT has a different tokenizer from our generation model, the Llama family, we integrate BERT with the Llama2 tokenizer and pre-train it for one epoch on 17GB Wikipedia documents (9 days using 8 of 24GB GPUs). Then, we finetune this BERT as TA, paired with the generation model of 1B (Tinyllama). This very lightweight TAALM is sufficiently trained on only a single 24GB GPU, significantly reducing resource use compared to the previous version (single 82GB GPU), thus making it affordable for the general environment.
>
> On the inference, the TA on BERT demonstrates compatibility with both the 1B and 7B generation models. Although its performance is below that of the TA on Llama, it still exhibits the highest performance among the other baselines. We will make sure to include this in the camera-ready version.
>
> - **Baselines with large generation model (Llama2 7B)**
> | | Parameter size of TA | Top Acc |  Epoch | NF Acc | Total Knowledge |
> |---|---|---|---|---|---|
> | Finetune | NA | 0.1150 | 16 | 0.8174 | 0.9324 |
> | TA (Llama) | 1.1B | 0.4290 | 4 | 0.8983 | 1.3273 |
> | TA (BERT) | 108M | 0.3210 | 6 | 0.9388 | 1.2598 |
>
> - **Baselines with small generation model (Tinyllama 1B)**
> | | Parameter size of TA | Top Acc | Epoch | NF Acc | Total Knowledge |
> |---|---|---|---|---|---|
> | Finetune | NA | 0.0700 | 29 | 0.7693 | 0.8393 |
> | TA (Llama) | 1.1B | 0.3260 | 4 | 0.9078 | 1.2338 |
> | TA (BERT) | 108M | 0.2440 | 9 | 0.9267 | 1.1707 |
> ***
> **[W3] Ablation Studies: While the paper includes comprehensive experiments, additional ablation studies to isolate the impact of different components of TAALM (e.g., the specific meta-learning algorithm used) would strengthen the evaluation.**
>
> Based on the token importance predicted by the TA, various design choices are possible. We explore and compare the effectiveness of these variations. This study enhances our understanding of how generative LM interacts with token weights when learning data. Detailed descriptions of the components and experimental results can be found in the **global rebuttal**.
> ***
>
> _**References**_
> [1] Sqil: Imitation learning via reinforcement learning with sparse rewards
> [2] Learning implicit credit assignment for cooperative multi-agent reinforcement learning

---

> > ### Comment · Reviewer_CkYa · 2024-08-09
> > **Response to Rebuttal  by Authors**
> >
> > Thanks for your detailed response to my questions. I am impressed with the results you achieved with both larger and smaller pre-trained models. I would like to keep my score.

---

> > > ### Author Response · Authors · 2024-08-12
> > > **Sincere thanks**
> > >
> > > We appreciate for the positive feedback on our response. Your review has strengthened our work. We will incorporate your feedback into the camera-ready version. We would be happy to address any further questions/suggestions that might come up until the end of the discussion period.

---

### Author Rebuttal · Authors · 2024-08-06

# Global Rebuttal


We first thank all reviewers for their thoughtful feedback on our work. We would like to address a suggestion commonly raised by reviewers, and introduce our progress in significantly reducing the GPU resource required.
We believe that constructive suggestions from all reviewers, such as the experiment on the token dropping method can significantly enhance the clarity and advancement of our paper.
we will make sure to include the updates in the camera-ready version
***
## **§A. Ablation study**
Based on the token importance predicted by the Train-Attention (TA), various design choices are possible. We explore and compare the effectiveness of these variations. This study enhances our understanding of how generative LM interacts with token weights when learning data.

The description of the components and experimental results are as follows:

1) Token-importance weight (ours) :
The original variation that utilizes token-importance weight predicted by TA for target-weighted learning.

2) Known token masking :
Masking out the tokens in real-time when prediction and label matches. This method is intended to enhance “model awareness” in TA, as TA is more oriented to “task awareness.”

3) Token weight dropping :
Among token-weight generated by TA, dropping weights that are below the top k% levels. We tested 50% and 80%. Vanilla TA is the same as the threshold of 0%. This method is intended to cut out noisy targets, as TA is supposed to assign lower weight to un-useful tokens. (suggested by reviewer **6R4i**)

### **Result**:
Known token masking does not yield better results compared to TAALM w/ token-importance weight. We hypothesize that the effect of known masking is limited because task awareness is already achieved when the loss of learned tokens is reduced.


Test results on the TAALM w/ token weight dropping show that as the threshold increases, the top accuracy decreases. This suggests that some useful targets are mixed in among the lower weights, and it helps the model learn better somehow. On the contrary, not-to-forget accuracy slightly improves as the threshold increases. This seems as the effect of 1) cutting out noisy targets and 2) trade-off for lower learning. However, Total Knowledge is best on the TAALM w/ token-importance weight (ours).

Overall experimental results indicate that, since TA is optimized to maximize task performance, adding heuristic interventions appears to produce suboptimal outcomes.

- **Baselines with large generation model (Llama2 7B)**

| | Top Acc | Epoch | NF Acc | Total Knowledge |
|---|---|---|---|---|
| Finetune | 0.1150 | 16 | 0.8174 | 0.9324 |
| TAALM w/ token-importance weight (**ours**) | 0.4290 | 4 | 0.8983 | 1.3273 |
| TAALM w/ known token masking  | 0.3920 | 4 | 0.9075 | 1.2995 |
| TAALM w/ token weight dropping < 0.5 | 0.4100 | 7 | 0.9148 | 1.3248 |
| TAALM w/ token weight dropping < 0.8 | 0.3850 | 4 | 0.9267 | 1.3117 |

***
## **§B. Reduction of resources through TA on BERT**
Due to the substantial GPU resources required to train the TA, we endeavored to find ways to reduce resource consumption.
A promising approach is utilizing Bidirectional Transformer (BERT) as a body for TA, which has high inferential capabilities even at a very small size (108M) compared to the previous body (Tinyllama 1.1B), due to its bidirectional property.
Since BERT has a different tokenizer from our generation model, the Llama family, we integrate BERT with the Llama2 tokenizer and pre-train it for one epoch on 17GB Wikipedia documents (9 days using 8 of 24GB GPUs). Then, we finetune this BERT as TA, paired with the generation model of 1B (Tinyllama). This very lightweight TAALM is sufficiently trained on only a single 24GB GPU, significantly reducing resource use compared to the previous version (single 82GB GPU), thus making it affordable for the general environment.
On the inference, the TA on BERT demonstrates compatibility with both the 1B and 7B generation models. Although its performance is below that of the TA on Llama, it still exhibits the highest performance among the other baselines. We will make sure to include this in the camera-ready version.

- **Baselines with large generation model (Llama2 7B)**
| | Parameter size of TA | Top Acc | Epoch | NF Acc | Total Knowledge |
|---|---|---|---|---|---|
| Finetune | NA | 0.1150 | 16 | 0.8174 | 0.9324 |
| TA (Llama) | 1.1B | 0.4290 | 4 | 0.8983 | 1.3273 |
| **TA (BERT)** | 108M | 0.3210 | 6 | 0.9388 | 1.2598 |


- **Baselines with small generation model (Tinyllama 1B)**
| | Parameter size of TA | Top Acc | Epoch | NF Acc | Total Knowledge |
|---|---|---|---|---|---|
| Finetune | NA | 0.0700 | 29 | 0.7693 | 0.8393 |
| TA (Llama) | 1.1B | 0.3260 | 4 | 0.9078 | 1.2338 |
| **TA (BERT)** | 108M | 0.2440 | 9 | 0.9267 | 1.1707 |

---

### Comment · Area_Chair_Xf9o · 2024-08-11
**Discussion**

Dear Reviewers,

Thanks for the reviews. The authors have uploaded their responses to your comments, please check if the rebuttal address your concerns and if you have further questions/comments to discuss with the authors. If the authors have addressed your concerns, please adjust your rating accordingly or vice versa.

Regards,

AC

---

### Decision · Program_Chairs · 2024-09-25

**Decision:**

Accept (poster)

**Comment:**

This paper investigates how to improve training efficiency of continual knowledge learning in large language models. It presents a meta-learning approach to estimating the importance/weights of tokens, reducing unnecessary parameter updating, and mitigating catastrophic forgetting. It further proposes a new benchmark for evaluating the trade-off between learning and retaining. Experiments demonstrate the effectiveness of the proposed method.

Code and datasets are provided for reproducing the experimental results.

All reviewers found this paper provides novel approach and new benchmark for continual knowledge learning which would be significant contributions to the ML community. After the rebuttal and discussion, all reviewers keep their positive scores. I recommend acceptance.